# Universal Successor Features Approximators

**Diana Borsa**[*]**, André Barreto, John Quan, Daniel Mankowitz, Rémi Munos**
**Hado van Hasselt, David Silver, Tom Schaul**
DeepMind
London, UK
{borsa,andrebarreto,johnquan,dmankowitz,munos,
hado,davidsilver,schaul}@google.com

## Abstract

The ability of a reinforcement learning (RL) agent to learn about many reward functions at the same time has many potential benefits, such as the decomposition of complex tasks into simpler ones, the exchange of information between tasks, and the reuse of skills. We focus on one aspect in particular, namely the ability to generalise to unseen tasks. Parametric generalisation relies on the interpolation power of a function approximator that is given the task description as input; one of its most common form are universal value function approximators (UVFAs). Another way to generalise to new tasks is to exploit structure in the RL problem itself. Generalised policy improvement (GPI) combines solutions of previous tasks into a policy for the unseen task; this relies on instantaneous policy evaluation of old policies under the new reward function, which is made possible through successor features (SFs). Our proposed *universal successor features approximators* (USFAs) combine the advantages of all of these, namely the scalability of UVFAs, the instant inference of SFs, and the strong generalisation of GPI. We discuss the challenges involved in training a USFA, its generalisation properties and demonstrate its practical benefits and transfer abilities on a large-scale domain in which the agent has to navigate in a first-person perspective three-dimensional environment.

## 1 Introduction

Reinforcement learning (RL) provides a general framework to model sequential decision-making problems with sparse evaluative feedback in the form of rewards. The recent successes in deep RL have prompted interest in increasingly more complex tasks and a shift in focus towards scenarios in which a single agent must solve multiple related problems, either simultaneously or sequentially. This paradigm is formally known as *multitask RL* (Taylor and Stone, 2009; Teh et al., 2017). One of the benefits of learning about multiple tasks at the same time is the possibility of *transferring* knowledge across tasks; in essence, this means that by jointly learning about a set of tasks one should be able to exploit their common structure to speed up learning and induce better generalisation (Taylor and Stone, 2009; Lazaric, 2012). A particularly interesting instance of transfer is the generalisation to new, unseen tasks. This potentially allows an agent to perform a task with little or no learning by leveraging knowledge from previously learned tasks. In this paper we will be exploring this scenario.

Consider an RL agent in a persistent environment trying to master a number of tasks. In order to generalise to unseen tasks, the agent needs to be able to identify and exploit some common structure underlying the tasks. Two possible sources of structure in this scenario are: i) some similarity between the solutions of the tasks, either in the policy or in the associated value-function space, and ii) the shared dynamics of the environment (*e.g.*, physics). In this paper we will attempt to build an agent that can make use of both types of structure. For this, we will build on two frameworks that exploit these structures in isolation. The first one are Schaul et al.'s (2015) *universal value function approximators* (UVFAs). UVFAs extend the notion of value functions to also include the description of a task, thus directly exploiting the common structure in the associated optimal value functions. The second framework we build upon exploits the common structure in the environment and capitalises on the power of dynamic programming. Barreto et al.'s (2017) framework is based on two core

---

[*]Corresponding author: borsa@google.com

concepts: *successor features* (SFs), a representation scheme that allows a policy to be evaluated on any task of a given format, and *generalised policy improvement* (GPI), a generalisation of dynamic programming's classic operator that uses a set of policies instead of a single one.

UVFAs and SF&GPI generalise to new tasks in quite different, and potentially complementary, ways. UVFAs aim to generalise across the space of tasks by exploiting structure in the underlying space of value functions. In contrast, SF&GPI strategy is to exploit the structure of the RL problem itself. In this paper we propose a model that exhibits both types of generalisation. The basic insight is to note that SFs are multi-dimensional value functions, so we can extend them in the same way a universal value function extends their unidimensional counterparts. We call the resulting model *universal successor features approximators*, or USFAs for short. USFA is a strict generalisation of its precursors. Specifically, we show that by combining USFAs and GPI we can recover both UVFAs and SF&GPI as particular cases. This opens up a new spectrum of possible approaches in between these two extreme cases. We discuss the challenges involved in training a USFA and demonstrate the practical benefits of doing so on a large-scale domain in which the agent has to navigate in a three-dimensional environment using only images as observations.

## 2 BACKGROUND

In this section we present some background material, formalise the scenario we are interested in, and briefly describe the methods we build upon.

### 2.1 MULTITASK REINFORCEMENT LEARNING

We consider the usual RL framework: an agent interacts with an environment and selects actions in order to maximise the expected amount of reward received in the long run (Sutton and Barto, 1998). As usual, we assume that such an interaction can be modeled as a *Markov decision process* (MDP, Puterman, 1994). An MDP is defined as a tuple $M \equiv (\mathcal{S}, \mathcal{A}, p, R, \gamma)$ where $\mathcal{S}$ and $\mathcal{A}$ are the state and action spaces, $p(\cdot|s, a)$ gives the next-state distribution upon taking action $a$ in state $s$, $R(s, a, s')$ is a random variable representing the reward received at transition $s \xrightarrow{a} s'$, and $\gamma \in [0, 1)$ is a discount factor that gives smaller weights to future rewards.

As mentioned in the introduction, in this paper we are interested in the multitask RL scenario, where the agent has to solve multiple *tasks*. Each task is defined by a reward function $R_\mathbf{w}$; thus, instead of a single MDP $M$, our environment is a *set* of MDPs that share the same structure except for the reward function. Following Barreto et al. (2017), we assume that the expected one-step reward associated with transition $s \xrightarrow{a} s'$ is given by

$$\mathrm{E}\left[R_\mathbf{w}(s, a, s')\right] = r_\mathbf{w}(s, a, s') = \boldsymbol{\phi}(s, a, s')^\top \mathbf{w}, \tag{1}$$

where $\boldsymbol{\phi}(s, a, s') \in \mathbb{R}^d$ are features of $(s, a, s')$ and $\mathbf{w} \in \mathbb{R}^d$ are weights. The features $\boldsymbol{\phi}(s, a, s')$ can be thought of as salient events that may be desirable or undesirable to the agent, such as for example picking up an object, going through a door, or knocking into something. In this paper we assume that the agent is able to recognise such events—that is, $\boldsymbol{\phi}$ is observable—, but the solution we propose can be easily extended to the case where $\boldsymbol{\phi}$ must be learned (Barreto et al., 2017). Our solution also applies to the case where (1) is only approximately satisfied, as discussed by Barreto et al. (2018).

Given $\mathbf{w} \in \mathbb{R}^d$ representing a task, the goal of the agent is to find a *policy* $\pi_\mathbf{w} : \mathcal{S} \mapsto \mathcal{A}$ that maximises the expected discounted sum of rewards, also called the *return* $G_\mathbf{w}^{(t)} = \sum_{i=0}^{\infty} \gamma^i R_\mathbf{w}^{(t+i)}$, where $R_\mathbf{w}^{(t)} = R_\mathbf{w}(S_t, A_t, S_{t+1})$ is the reward received at the $t^\text{th}$ time step. A principled way to address this problem is to use methods derived from *dynamic programming* (DP), which heavily rely on the concept of a *value function* (Puterman, 1994). The *action-value function* of a policy $\pi$ on task $\mathbf{w}$ is defined as $Q_\mathbf{w}^\pi(s, a) \equiv \mathrm{E}^\pi \left[G_\mathbf{w}^{(t)} \mid S_t = s, A_t = a\right]$, where $\mathrm{E}^\pi[\cdot]$ denotes expected value when following policy $\pi$. Based on $Q_\mathbf{w}^\pi$ we can compute a *greedy* policy $\pi'(s) \in \mathrm{argmax}_a Q_\mathbf{w}^\pi(s, a)$; one of the fundamental results in DP guarantees that $Q_\mathbf{w}^{\pi'}(s, a) \geq Q_\mathbf{w}^\pi(s, a)$ for all $(s, a) \in \mathcal{S} \times \mathcal{A}$. The computation of $Q_\mathbf{w}^\pi(s, a)$ and $\pi'$ are called *policy evaluation* and *policy improvement*; under certain conditions their successive application leads to the optimal value function $Q_\mathbf{w}^*$, from which one can derive an optimal policy for task $\mathbf{w}$ as $\pi_\mathbf{w}^*(s) \in \mathrm{argmax}_a Q_\mathbf{w}^*(s, a)$ (Sutton and Barto, 1998).

As a convention, in this paper we will add a tilde to a symbol to indicate that the associated quantity is an approximation; we will then refer to the respective tunable parameters as $\boldsymbol{\theta}$. For example, the agent computes an approximation $\tilde{Q}_{\mathbf{w}}^{\pi} \approx Q_{\mathbf{w}}^{\pi}$ by tuning $\boldsymbol{\theta}_Q$.

## 2.2 TRANSFER LEARNING

Here we focus on one aspect of multitask RL: how to *transfer* knowledge to unseen tasks (Taylor and Stone, 2009; Lazaric, 2012). Specifically, we ask the following question: how can an agent leverage knowledge accumulated on a set of tasks $\mathcal{M} \subset \mathbb{R}^d$ to speed up the solution of a new task $\mathbf{w}' \notin \mathcal{M}$?

In order to investigate the question above we recast it using the formalism commonly adopted in learning. Specifically, we define a distribution $\mathcal{D}_{\mathbf{w}}$ over $\mathbb{R}^d$ and assume the goal is for the agent to perform as well as possible under this distribution. As usual, we assume a fixed budget of sample transitions and define a training set $\mathcal{M} \sim \mathcal{D}_{\mathbf{w}}$ that is used by the agent to learn about the tasks of interest. We also define a test set $\mathcal{M}' \sim \mathcal{D}_{\mathbf{w}}$ and use it to assess the agent's generalisation—that is, how well it performs on the distribution of MDPs induced by $\mathcal{D}_{\mathbf{w}}$.

A natural way to address the learning problem above is to use Schaul et al.'s (2015) *universal value-function approximators* (UVFAs). The basic insight behind UVFAs is to note that the concept of optimal value function can be extended to include as one of its arguments a description of the task; an obvious way to do so in the current context is to define the function $Q^*(s, a, \mathbf{w}) : \mathcal{S} \times \mathcal{A} \times \mathbb{R}^d \mapsto \mathbb{R}$ as the optimal value function associated with task $\mathbf{w}$. The function $Q^*(s, a, \mathbf{w})$ is called a *universal value function* (UVF); a UVFA is then the corresponding approximation, $\tilde{Q}(s, a, \mathbf{w})$. When we define transfer as above it becomes clear that in principle a sufficiently expressive UVFA can identify and exploit structure across the joint space $\mathcal{S} \times \mathcal{A} \times \mathbb{R}^d$. In other words, a properly trained UVFA should be able to generalise across the space of tasks.

A different way of generalising across tasks is to use Barreto et al.'s (2017) framework, which builds on assumption (1) and two core concepts: *successor features* (SFs) and *generalised policy improvement* (GPI). The SFs $\boldsymbol{\psi} \in \mathbb{R}^d$ of a state-action pair $(s, a)$ under policy $\pi$ are given by

$$\boldsymbol{\psi}^{\pi}(s, a) \equiv \mathrm{E}^{\pi}\left[\sum_{i=t}^{\infty} \gamma^{i-t} \boldsymbol{\phi}_{i+1} \,|\, S_t = s, A_t = a\right]. \tag{2}$$

SFs allow one to immediately compute the value of a policy $\pi$ on *any* task $\mathbf{w}$: it is easy to show that, when (1) holds, $Q_{\mathbf{w}}^{\pi}(s, a) = \boldsymbol{\psi}^{\pi}(s, a)^{\top}\mathbf{w}$. It is also easy to see that SFs satisfy a Bellman equation in which $\boldsymbol{\phi}$ play the role of rewards, so $\boldsymbol{\psi}$ can be learned using any RL method (Szepesvári, 2010).

GPI is a generalisation of the policy improvement step described in Section 2.1. The difference is that in GPI the improved policy is computed based on a *set* of value functions rather than on a single one. Suppose that the agent has learned the SFs $\boldsymbol{\psi}^{\pi_i}$ of policies $\pi_1, \pi_2, ..., \pi_n$. When exposed to a new task defined by $\mathbf{w}$, the agent can immediately compute $Q_{\mathbf{w}}^{\pi_i}(s, a) = \boldsymbol{\psi}^{\pi_i}(s, a)^{\top}\mathbf{w}$. Let the GPI policy be defined as $\pi(s) \in \operatorname{argmax}_a Q^{\max}(s, a)$, where $Q^{\max} = \max_i Q^{\pi_i}$. The GPI theorem states that $Q^{\pi}(s, a) \geq Q^{\max}(s, a)$ for all $(s, a) \in \mathcal{S} \times \mathcal{A}$. The result also extends to the scenario where we replace $Q^{\pi_i}$ with approximations $\tilde{Q}^{\pi_i}$ (Barreto et al., 2017).

# 3 UNIVERSAL SUCCESSOR FEATURES APPROXIMATORS

UVFAs and SF&GPI address the transfer problem described in Section 2.2 in quite different ways. With UVFAs, one trains an approximator $\tilde{Q}(s, a, \mathbf{w})$ by solving the training tasks $\mathbf{w} \in \mathcal{M}$ using any RL algorithm of choice. One can then generalise to a new task by plugging its description $\mathbf{w}'$ into $\tilde{Q}$ and then acting according to the policy $\pi(s) \in \operatorname{argmax}_a \tilde{Q}(s, a, \mathbf{w}')$. With SF&GPI one solves each task $\mathbf{w} \in \mathcal{M}$ and computes an approximation of the SFs of the resulting policies $\pi_{\mathbf{w}}$, $\tilde{\boldsymbol{\psi}}^{\pi_{\mathbf{w}}}(s, a) \approx \boldsymbol{\psi}^{\pi_{\mathbf{w}}}(s, a)$. The way to generalise to a new task $\mathbf{w}'$ is to use the GPI policy defined as $\pi(s) \in \operatorname{argmax}_a \max_{\mathbf{w} \in \mathcal{M}} \tilde{\boldsymbol{\psi}}^{\pi_{\mathbf{w}}}(s, a)^{\top}\mathbf{w}'$.

The algorithmic differences between UVFAs and SF&GPI reflect the fact that these approaches exploit distinct properties of the transfer problem. UVFAs aim at generalising across the space of tasks by exploiting structure in the function $Q^*(s, a, \mathbf{w})$. In practice, such strategy materialises in the choice of function approximator, which carries assumptions about the shape of $Q^*(s, a, \mathbf{w})$. For example, by using a neural network to represent $\tilde{Q}(s, a, \mathbf{w})$ one is implicitly assuming that

$Q^*(s, a, \mathbf{w})$ is smooth in the space of tasks; roughly speaking, this means that small perturbations to $\mathbf{w}$ will result in small changes in $Q^*(s, a, \mathbf{w})$.

In contrast, SF&GPI's strategy to generalise across tasks is to exploit structure in the RL problem itself. GPI builds on the general fact that a greedy policy with respect to a value function will in general perform better than the policy that originated the value function. SFs, in turn, exploit the structure (1) to make it possible to quickly evaluate policies across tasks—and thus to apply GPI in an efficient way. The difference between the types of generalisation promoted by UVFAs and SF&GPI is perhaps even clearer when we note that the latter is completely agnostic to the way the approximations $\tilde{\psi}^{\pi_{\mathbf{w}}}$ are represented, and in fact it can applied even with a tabular representation.

Obviously, both UVFAs and GPI have advantages and limitations. In order to illustrate this point, consider two tasks $\mathbf{w}$ and $\mathbf{w}'$ that are "similar", in the sense that $||\mathbf{w} - \mathbf{w}'||$ is small ($|| \cdot ||$ is a norm in $\mathbb{R}^d$). Suppose that we have trained a UVFA on task $\mathbf{w}$ and as a result we obtained a good approximation $\tilde{Q}(s, a, \mathbf{w}) \approx Q^*(s, a, \mathbf{w})$. If the structural assumptions underlying $\tilde{Q}(s, a, \mathbf{w})$ hold— for example, $\tilde{Q}(s, a, \mathbf{w})$ is smooth with respect to $\mathbf{w}$—, it is likely that $\tilde{Q}(s, a, \mathbf{w}')$ will be a good approximation of $Q^*(s, a, \mathbf{w}')$. On the other hand, if such assumptions do not hold, we should not expect UVFA to perform well. A sufficient condition for SF&GPI to generalise well from task $\mathbf{w}$ to task $\mathbf{w}'$ is that the policy $\pi(s) \leftarrow \operatorname{argmax}_a \tilde{\psi}^{\pi_{\mathbf{w}}}(s, a)^\top \mathbf{w}'$ does well on task $\mathbf{w}'$, where $\pi_{\mathbf{w}}$ is a solution for task $\mathbf{w}$. On the downside, SF&GPI will not exploit functional regularities at all, even if they do exist. Let policy $\pi_{\mathbf{w}'}$ be a solution for tasks $\mathbf{w}'$. In principle we cannot say anything about $\psi^{\pi_{\mathbf{w}'}}(s, a)$, the SFs of $\pi_{\mathbf{w}'}$, even if we have a good approximation $\tilde{\psi}^{\pi_{\mathbf{w}}}(s, a) \approx \psi^{\pi_{\mathbf{w}}}(s, a)$.

As one can see, the types of generalisation provided by UVFAs and SF&GPI are in some sense complementary. It is then natural to ask if we can simultaneously have the two types of generalisation. In this paper we propose a model that provides exactly that. The main insight is actually simple: since SFs are multi-dimensional value functions, we can extend them in the same way as universal value functions extend regular value functions. In the next section we elaborate on how exactly to do so.

## 3.1 UNIVERSAL SUCCESSOR FEATURES

As discussed in Section 2.2, UVFs are an extension of standard value functions defined as $Q^*(s, a, \mathbf{w})$. If $\pi_{\mathbf{w}}$ is one of the optimal policies of task $\mathbf{w}$, we can rewrite the definition as $Q^{\pi_{\mathbf{w}}}(s, a, \mathbf{w})$. This makes it clear that the argument $\mathbf{w}$ plays two roles in the definition of a UVF: it determines both the task $\mathbf{w}$ and the policy $\pi_{\mathbf{w}}$ (which will be optimal with respect to $\mathbf{w}$). This does not have to be the case, though. Similarly to Sutton et al.'s (2011) *general value functions* (GVFs), we could in principle define a function $Q(s, a, \mathbf{w}, \pi)$ that "disentangles" the task from the policy. This would provide a model that is even more general than UVFs. In this section we show one way to construct such a model when assumption (1) (approximately) holds.

Note that, when (1) is true, we can revisit the definition of SFs and write $Q(s, a, \mathbf{w}, \pi) = \psi^\pi(s, a)^\top \mathbf{w}$. If we want to be able to compute $Q(s, a, \mathbf{w}, \pi)$ for any $\pi$, we need SFs to span the space of policies $\pi$. Thus, we define *universal successor features* as $\psi(s, a, \pi) \equiv \psi^\pi(s, a)$. Based on such definition, we call $\tilde{\psi}(s, a, \pi) \approx \psi(s, a, \pi)$ a *universal successor features approximator* (USFA).

In practice, when defining a USFA we need to define a representation for the policies $\pi$. A natural choice is to embed $\pi$ onto $\mathbb{R}^k$. Let $e : (\mathcal{S} \mapsto \mathcal{A}) \mapsto \mathbb{R}^k$ be a *policy-encoding mapping*, that is, a function that turns policies $\pi$ into vectors in $\mathbb{R}^k$. We can then see USFs as a function of $e(\pi)$: $\psi(s, a, e(\pi))$. The definition of the policy-encoding mapping $e(\pi)$ can have a strong impact on the structure of the resulting USF. We now point out a general equivalence between policies and reward functions that will provide a practical way of defining $e(\pi)$. It is well known that any reward function induces a set of optimal policies (Puterman, 1994). A point that is perhaps less immediate is that the converse is also true. Given a deterministic policy $\pi$, one can easily define a set of reward functions that induce this policy: for example, we can have $r_\pi(s, \pi(s), \cdot) = 0$ and $r_\pi(s, a, \cdot) = c$, with $c < 0$, for any $a \neq \pi(s)$. Therefore, we can use rewards to refer to deterministic policies and vice-versa (as long as potential ambiguities are removed when relevant).

Since here we are interested in reward functions of the form (1), if we restrict our attention to policies induced by tasks $\mathbf{z} \in \mathbb{R}^d$ we end up with a conveniently simple encoding function $e(\pi_{\mathbf{z}}) = \mathbf{z}$. From this encoding function it follows that $Q(s, a, \mathbf{w}, \pi_{\mathbf{z}}) = Q(s, a, \mathbf{w}, \mathbf{z})$. It should be clear that UVFs are a particular case of this definition when $\mathbf{w} = \mathbf{z}$. Going back to the definition of USFs, we can

finally write $Q(s, a, \mathbf{w}, \mathbf{z}) = \boldsymbol{\psi}(s, a, \mathbf{z})^\top \mathbf{w}$. Thus, if we learn a USF $\boldsymbol{\psi}(s, a, \mathbf{z})$, we have a value function that generalises over both tasks and policies, as promised.

## 3.2 USFA GENERALISATION

We now revisit the question as to why USFAs should provide the benefits associated with both UVFAs and SF&GPI. We will discuss how exactly to train a USFA in the next section, but for now suppose that we have trained one such model $\tilde{\boldsymbol{\psi}}(s, a, \mathbf{z})$ using the training tasks in $\mathcal{M}$. It is then not difficult to see that we can recover the solutions provided by both UVFAs and SF&GPI. Given an unseen task $\mathbf{w}'$, let $\pi$ be the GPI policy defined as

$$\pi(s) \in \operatorname{argmax}_a \max_{\mathbf{z} \in \mathcal{C}} \tilde{Q}(s, a, \mathbf{w}', \mathbf{z}) = \operatorname{argmax}_a \max_{\mathbf{z} \in \mathcal{C}} \tilde{\boldsymbol{\psi}}(s, a, \mathbf{z})^\top \mathbf{w}', \tag{3}$$

where $\mathcal{C} \subset \mathbb{R}^d$. Clearly, if we make $\mathcal{C} = \{\mathbf{w}'\}$, we get exactly the sort of generalisation associated with UVFAs. On the other hand, setting $\mathcal{C} = \mathcal{M}$ essentially recovers SF&GPI.

The fact that we can recover both UVFAs and SF&GPI opens up a spectrum of possibilities in between the two. For example, we could apply GPI over the training set augmented with the current task, $\mathcal{C} = \mathcal{M} \cup \{\mathbf{w}'\}$. In fact, USFAs allow us to apply GPI over *any* set of tasks $\mathcal{C} \subset \mathbb{R}^d$. The benefits of this flexibility are clear when we look at the theory supporting SF&GPI, as we do next.

Barreto et al. (2017) provide theoretical guarantees on the performance of SF&GPI applied to any task $\mathbf{w}' \in \mathcal{M}'$ based on a fixed set of SFs. Below we state a slightly more general version of this result that highlights the two types of generalisation promoted by USFAs (proof in Barreto et al.'s (2017) Theorem 2).

**Proposition 1** *Let $\mathbf{w}' \in \mathcal{M}'$ and let $Q_{\mathbf{w}'}^\pi$ be the action-value function of executing policy $\pi$ on task $\mathbf{w}'$. Given approximations $\{\tilde{Q}_{\mathbf{w}'}^{\pi_\mathbf{z}} = \tilde{\boldsymbol{\psi}}(s, a, \mathbf{z})^\top \mathbf{w}'\}_{\mathbf{z} \in \mathcal{C}}$, let $\pi$ be the GPI policy defined in 3. Then,*

$$\|Q_{\mathbf{w}'}^* - Q_{\mathbf{w}'}^\pi\|_\infty \leq \frac{2}{1-\gamma} \min_{\mathbf{z} \in \mathcal{C}} \left( \|\boldsymbol{\phi}\|_\infty \underbrace{\|\mathbf{w}' - \mathbf{z}\|}_{\delta_d(\mathbf{z})} \right) + \max_{\mathbf{z} \in \mathcal{C}} \left( \|\mathbf{w}'\| \cdot \underbrace{\|\boldsymbol{\psi}^{\pi_\mathbf{z}} - \tilde{\boldsymbol{\psi}}(s, a, \mathbf{z})\|_\infty}_{\delta_\psi(\mathbf{z})} \right), \tag{4}$$

*where $Q_{\mathbf{w}'}^*$ is the optimal value of task $\mathbf{w}'$, $\psi^{\pi_\mathbf{z}}$ are the SFs corresponding to the optimal policy for task $\mathbf{z}$, and $\|f - g\|_\infty = \max_{s,a} |f(s, a) - g(s, a)|$.*

When we write the result in this form, it becomes clear that, for each policy $\pi_\mathbf{z}$, the right-hand side of (4) involves two terms: i) $\delta_d(\mathbf{z})$, the distance between the task of interest $\mathbf{w}'$ and the task that induced $\pi_\mathbf{z}$, and ii) $\delta_\psi(\mathbf{z})$, the quality of the approximation of the SFs associated with $\pi_\mathbf{z}$.

In order to get the tightest possible bound (4) we want to include in $\mathcal{C}$ the policy $\mathbf{z}$ that minimises $\delta_d(\mathbf{z}) + \delta_\psi(\mathbf{z})$. This is exactly where the flexibility of choosing $\mathcal{C}$ provided by USFAs can come in handy. Note that, if we choose $\mathcal{C} = \mathcal{M}$, we recover Barreto et al.'s (2017) bound, which may have an irreducible $\min_{\mathbf{z} \in \mathcal{C}} \delta_d(\mathbf{z})$ even with a perfect approximation of the SFs in $\mathcal{M}$. On the other extreme, we could query our USFA at the test point $\mathcal{C} = \{\mathbf{w}'\}$. This would result in $\delta_d(\mathbf{w}') = 0$, but can potentially incur a high cost due to the associated approximation error $\delta_\psi(\mathbf{w}')$.

## 3.3 HOW TO TRAIN A USFA

Now that we have an approximation $\tilde{Q}(s, a, \mathbf{w}, \mathbf{z})$ a natural question is how to train this model. In this section we show that the decoupled nature of $\tilde{Q}$ is reflected in the training process, which assigns clearly distinct roles for tasks $\mathbf{w}$ and policies $\pi_\mathbf{z}$.

In our scenario, a transition at time $t$ will be $(s_t, a_t, \boldsymbol{\phi}_{t+1}, s_{t+1})$. Note that $\boldsymbol{\phi}$ allows us to compute the reward for any task $\mathbf{w}$, and since our policies are encoded by $\mathbf{z}$, transitions of this form allow us to learn the value function of *any* policy $\pi_\mathbf{z}$ on *any* task $\mathbf{w}$. To see why this is so, let us define the temporal-difference (TD) error (Sutton and Barto, 1998) used in learning these value functions.

Given transitions in the form above, the $n$-step TD error associated with policy $\pi_{\mathbf{z}}$ on task $\mathbf{w}$ will be

$$
\boldsymbol{\delta}_{\mathbf{wz}}^{t,n} = \sum_{i=t}^{t+n-1} \gamma^{i-t} r_{\mathbf{w}}(s_i, a_i, s_{i+1}) + \gamma^n \tilde{Q}(s_{t+n}, \pi_{\mathbf{z}}(s_{t+n}), \mathbf{w}, \mathbf{z}) - \tilde{Q}(s_t, a_t, \mathbf{w}, \mathbf{z})
$$

$$
= \left[ \sum_{i=t}^{t+n-1} \gamma^{i-t} \phi(s_i, a_i, s_{i+1}) + \gamma^n \tilde{\psi}(s_{t+n}, a_{t+n}, \mathbf{z}) - \tilde{\psi}(s_t, a_t, \mathbf{z}) \right]^{\top} \mathbf{w} = (\boldsymbol{\delta}_{\mathbf{z}}^{t,n})^{\top} \mathbf{w}, \quad (5)
$$

where $a_{t+n} = \operatorname{argmax}_b \tilde{Q}(s_{t+n}, b, \mathbf{z}, \mathbf{z}) = \operatorname{argmax}_b \tilde{\psi}(s_{t+n}, b, \mathbf{z})^{\top} \mathbf{z}$. As is well known, the TD error $\boldsymbol{\delta}_{\mathbf{wz}}^{t,n}$ allows us to learn the value of policy $\pi_{\mathbf{z}}$ on task $\mathbf{w}$; since here $\boldsymbol{\delta}_{\mathbf{wz}}^{t,n}$ is a function of $\mathbf{z}$ and $\mathbf{w}$ only, we can learn about any policy $\pi_{\mathbf{z}}$ on any task $\mathbf{w}$ by just plugging in the appropriate vectors.

Equation (5) highlights some interesting (and subtle) aspects involved in training a USFA. Since the value function $\tilde{Q}(s, a, \mathbf{w}, \mathbf{z})$ can be decoupled into two components, $\tilde{\psi}(s, a, \mathbf{z})$ and $\mathbf{w}$, the process of evaluating a policy on a task reduces to learning $\tilde{\psi}(s, a, \mathbf{z})$ using the vector-based TD error $\boldsymbol{\delta}_{\mathbf{z}}^{t,n}$ showing up in (5). Since $\boldsymbol{\delta}_{\mathbf{z}}^{t,n}$ is a function of $\mathbf{z}$ only, the updates to $\tilde{Q}(s, a, \mathbf{w}, \mathbf{z})$ will *not* depend on $\mathbf{w}$. How do the tasks $\mathbf{w}$ influence the training of a USFA, then? If sample transitions are collected by a behaviour policy, as is usually the case in online RL, a natural choice is to have this policy be induced by a task $\mathbf{w}$. When this is the case the training tasks $\mathbf{w} \in \mathcal{M}$ will define the distribution used to collect sample transitions. Whenever we want to update $\psi(s, a, \mathbf{z})$ for a different $\mathbf{z}$ than the one used in generating the data, we find ourselves under the *off-policy* regime (Sutton and Barto, 1998).

Assuming that the behaviour policy is induced by the tasks $\mathbf{w} \in \mathcal{M}$, training a USFA involves two main decisions: how to sample tasks from $\mathcal{M}$ and how to sample policies $\pi_{\mathbf{z}}$ to be trained through (5) or some variant. As alluded to before, these decisions may have a big impact on the performance of the resulting USFA, and in particular on the trade-offs involved in the choice of the set of policies $\mathcal{C}$ used by the GPI policy (3). As a form of illustration, Algorithm 1 shows a possible regime to train a USFA based on particularly simple strategies to select tasks $\mathbf{w} \in \mathcal{M}$ and to sample policies $\mathbf{z} \in \mathbb{R}^d$. One aspect of Algorithm 1 worth calling attention to is the fact that the distribution $\mathcal{D}_{\mathbf{z}}$ used to select policies can depend on the current task $\mathbf{w}$. This allows one to focus on specific regions of the policy space; for example, one can sample policies using a Gaussian distribution centred around $\mathbf{w}$.

---

**Algorithm 1** Learn USFA with $\epsilon$-greedy $Q$-learning

**Require:** $\epsilon$, training tasks $\mathcal{M}$, distribution $\mathcal{D}_{\mathbf{z}}$ over $\mathbb{R}^d$, number of policies $n_{\mathbf{z}}$
1: select initial state $s \in \mathcal{S}$
2: **for** ns steps **do**
3:     sample $\mathbf{w}$ uniformly at random from $\mathcal{M}$
4:     {sample policies, possibly based on current task}
5:     **for** $i \leftarrow 1, 2, ..., n_{\mathbf{z}}$ **do** $\mathbf{z}_i \sim \mathcal{D}_{\mathbf{z}}(\cdot|\mathbf{w})$
6:     **if** Bernoulli($\epsilon$)=1 **then** $a \leftarrow \text{Uniform}(\mathcal{A})$
7:     **else** $a \leftarrow \operatorname{argmax}_b \max_i \tilde{\psi}(s, b, \mathbf{z}_i)^{\top} \mathbf{w}$ {GPI }
8:     Execute action $a$ and observe $\phi$ and $s'$
9:     **for** $i \leftarrow 1, 2, ..., n_{\mathbf{z}}$ **do** {Update $\tilde{\psi}$}
10:       $a' \leftarrow \operatorname{argmax}_b \tilde{\psi}(s, b, \mathbf{z}_i)^{\top} \mathbf{z}_i$ {$a' \equiv \pi_i(s')$}
11:       $\boldsymbol{\theta} \xleftarrow{\alpha} \left[ \phi + \gamma \tilde{\psi}(s', a', \mathbf{z}_i) - \tilde{\psi}(s, a, \mathbf{z}_i) \right] \nabla_{\boldsymbol{\theta}} \tilde{\psi}$
12:     $s \leftarrow s'$
13: **return** $\boldsymbol{\theta}$

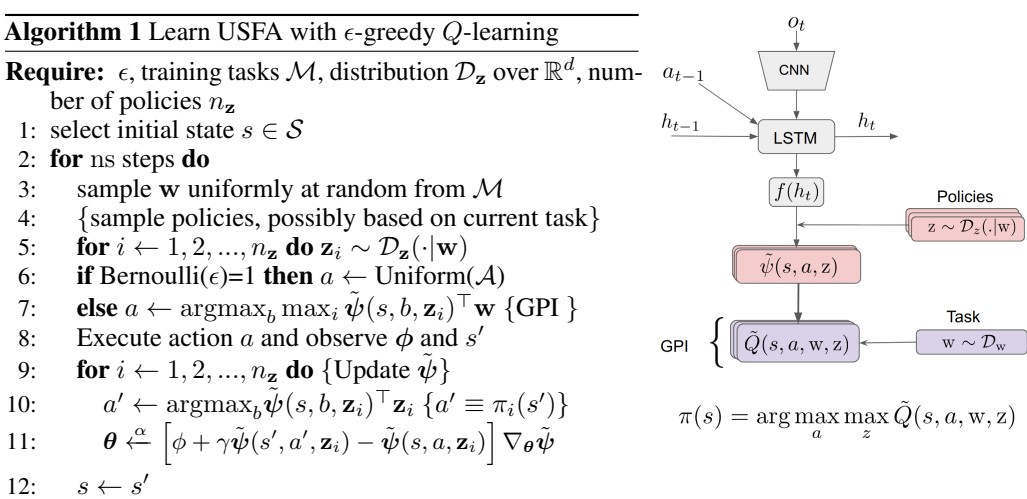

Figure 1: USFA architecture

# 4 EXPERIMENTS

In this section we describe the experiments conducted to test the proposed architecture in a multitask setting and assess its ability to generalise to unseen tasks.

## 4.1 ILLUSTRATIVE EXAMPLE: TRIP MDP

We start with a simple illustrative example to provide intuition on the kinds of generalisation provided by UVFAs and SF&GPI. We also show how in this example USFAs can effectively leverage both

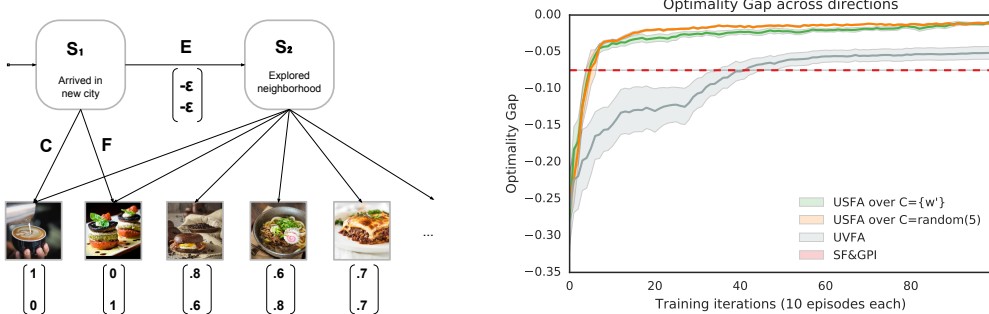

Figure 2: Trip MDP: [Left] Depiction of MDP. [Right] Optimality gap (Difference between optimal return and the return obtained by the different models) at different times in the training process.

types of generalisation and outperform its precursors. For this we will consider the simple two-state MDP depicted in Figure 2. To motivate the example, suppose that state $s_1$ of our MDP represents the arrival of a traveler to a new city. The traveler likes coffee and food and wants to try what the new city has to offer. In order to model that, we will use features $\phi \in \mathbb{R}^2$, with $\phi_1$ representing the quality of the coffee and $\phi_2$ representing the quality of the food, both ranging from $0$ to $1$. The traveler has done some research and identified the places that serve the best coffee and the best food; in our MDP these places are modelled by terminal states associated with actions '$C$' and '$F$' whose respective associated rewards are $\phi(\cdot, C) = \phi(C) = [1, 0]$ and $\phi(F) = [0, 1]$. As one can infer from these feature vectors, the best coffee place does not serve food and the best restaurant does not serve coffee (at least not a very good one). Nevertheless, there are other places in town that serve both; as before, we will model these places by actions $P_i$ associated with features $\phi(P_i)$. We assume that $\|\phi(P_i)\|_2 = 1$ and consider $N = 5$ alternative places $P_i$ evenly spaced on the preference spectrum. We model how much the traveler wants coffee and food on a given day by $\mathbf{w} \in \mathbb{R}^2$. If the traveler happens to want only one of these (i.e. $\mathbf{w} \in \{[1, 0], [0, 1]\}$), she can simply choose actions '$C$' or '$F$' and get a reward $r = \phi(\cdot)^\top \mathbf{w} = 1$. If instead she wants both coffee and food (i.e. if $\mathbf{w}$ is not an "one-hot" vector), it may actually be best to venture out to one of the other places. Unfortunately, this requires the traveler to spend some time researching the area, which we model by an action '$E$' associated with feature $\phi(E) = [-\epsilon, -\epsilon]$. After choosing '$E$' the traveler lands on state $s_2$ and can now reach any place in town: $C, F, P_1, ..., P_N$. Note that, depending on the vector of preferences $\mathbf{w}$, it may be worth paying the cost of $\phi(E)^\top \mathbf{w}$ to subsequently get a reward of $\phi(P_i)^\top \mathbf{w}$ (here $\gamma = 1$).

In order to assess the transfer ability of UVFAs, SF&GPI and USFAs, we define a training set $\mathcal{M} = \{10, 01\}$ and $K = 50$ test tasks corresponding to directions in the two-dimensional $\mathbf{w}$-space: $\mathcal{M}' = \left\{\mathbf{w}' | \mathbf{w}' = [\cos(\frac{\pi k}{2K}), \sin(\frac{\pi k}{2K})], k = 0, 1, ..., K\right\}$. We start by analysing what SF&GPI would do in this scenario. We focus on training task $\mathbf{w}_C = [1, 0]$, but an analogous reasoning applies to task $\mathbf{w}_F = [0, 1]$. Let $\pi_C$ be the optimal policy associated with task $\mathbf{w}_C$. It is easy to see that $\pi(s_1) = \pi(s_2) = C$. Thus, under $\pi_C$ it should be clear that $Q_{\mathbf{w}'}^{\pi_C}(s_1, C) > Q_{\mathbf{w}'}^{\pi_C}(s_1, E)$ for all test tasks $\mathbf{w}'$. Since the exact same reasoning applies to task $\mathbf{w}_F$ if we replace action $C$ with action $F$, the GPI policy (3) computed over $\{\boldsymbol{\psi}^{\pi_C}, \boldsymbol{\psi}^{\pi_F}\}$ will be suboptimal for most test tasks in $\mathcal{M}'$. Training a UVFA on the same set $\mathcal{M}$, will not be perfect either due to the very limited number of training tasks. Nonetheless the smoothness in the approximation allows for a slightly better generalisation in $\mathcal{M}'$.

Alternatively, we can use Algorithm 1 to train a USFA on the training set $\mathcal{M}$. In order to do so we sampled $n_{\mathbf{z}} = 5$ policies $\mathbf{z} \in \mathbb{R}^2$ using a uniformly random distribution $\mathcal{D}_{\mathbf{z}}(\cdot|\mathbf{w}) = \mathcal{U}([0, 1]^2)$ (see line 5 in Algorithm 1). When acting on the test tasks $\mathbf{w}'$ we considered two choices for the candidates set: $\mathcal{C} = \{\mathbf{w}'\}$ and $\mathcal{C} = \{\mathbf{z}_i | \mathbf{z}_i \sim \mathcal{U}([0, 1]^2), i = 1, 2, ..., 5\}$. Empirical results are provided in Figure 2. As a reference we report the performance of SF&GPI using the *true* $\{\boldsymbol{\psi}^{\pi_C}, \boldsymbol{\psi}^{\pi_F}\}$ – no approximation. We also show the learning curve of a UVFA. As shown in the figure, USFA clearly outperforms its precursors and quickly achieves near optimal performance. This is due to two factors. First, contrary to vanilla SF&GPI, USFA can discover and exploit the rich structure in the policy-space, enjoying the same generalisation properties as UVFAs but now enhanced by the combination of the off-policy *and* off-task training regime. Second, the ability to sample a candidate set $\mathcal{C}$ that induces some diversity in the policies considered by GPI overcomes the suboptimality associated with the training SFs $\boldsymbol{\psi}^{\pi_C}$ and $\boldsymbol{\psi}^{\pi_F}$. We explore this effect in a bit more detail in the suppl. material.

## 4.2 LARGE SCALE EXPERIMENTS

**Environment and tasks**. We used the DeepMind Lab platform to design a 3D environment consisting of one large room containing four types of objects: TVs, balls, hats, and balloons (Beattie et al., 2016; Barreto et al., 2018). A depiction of the environment through the eyes of the agent can be seen in Fig. 3. Features $\phi_i$ are indicator functions associated with object types, *i.e.*, $\phi_i(s, a, s') = 1$ if and only if the agent

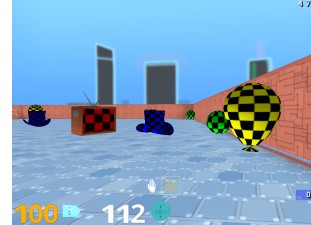

(a) Screenshot of environment

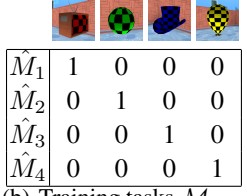

| | | | | |
|---|---|---|---|---|
| $\hat{M}_1$ | 1 | 0 | 0 | 0 |
| $\hat{M}_2$ | 0 | 1 | 0 | 0 |
| $\hat{M}_3$ | 0 | 0 | 1 | 0 |
| $\hat{M}_4$ | 0 | 0 | 0 | 1 |

(b) Training tasks $\mathcal{M}$

Figure 3: Environment.

collects an object of type $i$ (say, a TV) on the transition $s \xrightarrow{a} s'$. A task is defined by four real numbers $\mathbf{w} \in \mathbb{R}^4$ indicating the rewards attached to each object. Note that these numbers can be negative, in which case the agent has to avoid the corresponding object type. For instance, in task $\mathbf{w} = [1\text{-}100]$ the agent is interested in objects of the first type and should avoid objects of the second.

**Agent architecture**. A depiction of the architecture used for the USFA agent is illustrated in Fig. 1. The architecture has three main modules: i) A fairly standard *input processing unit* composed of three convolution layers and an LSTM followed by a non-linearity (Schmidhuber, 1996); ii) A *policy conditioning module* that combines the state embedding coming from the first module, $s \equiv f(h)$, and the policy embedding, $\mathbf{z}$, and produces $|A| \cdot d$ outputs corresponding to the SFs of policy $\pi_{\mathbf{z}}$, $\tilde{\psi}(s, a, \mathbf{z})$; and iii) The *evaluation module*, which, given a task $\mathbf{w}$ and the SFs $\tilde{\psi}(s, a, \mathbf{z})$, will construct the evaluation of policy $\pi_{\mathbf{z}}$ on $\mathbf{w}$, $\tilde{Q}(s, a, \mathbf{w}, \mathbf{z}) = \tilde{\psi}(s, a, \mathbf{z})^\top \mathbf{w}$.

**Training and baselines**. We trained the above architecture end-to-end using a variation of Alg. 1 that uses Watkins's (1989) $Q(\lambda)$ to apply $Q$-learning with eligibility traces. As for the distribution $\mathcal{D}_{\mathbf{z}}$ used in line 5 of Alg. 1 we adopted a Gaussian centred at $\mathbf{w}$: $\mathbf{z} \sim \mathcal{N}(\mathbf{w}, 0.1\,\mathbf{I})$, where $\mathbf{I}$ is the identity matrix. We used the canonical vectors of $\mathbb{R}^4$ as the training set, $\mathcal{M} = \{1000, 0100, 0010, 0001\}$. Once an agent was trained on $\mathcal{M}$ we evaluated it on a separate set of unseen tasks, $\mathcal{M}'$, using the GPI policy (3) over different sets of policies $\mathcal{C}$. Specifically, we used: $\mathcal{C} = \{\mathbf{w}'\}$, which corresponds to a UVFA with an architecture specialised to (1); $\mathcal{C} = \mathcal{M}$, which corresponds to doing GPI on the SFs of the training policies (similar to (Barreto et al., 2017)), and $\mathcal{C} = \mathcal{M} \cup \{\mathbf{w}'\}$, which is a combination of the previous two. We also included as baselines two standard UVFAs that do not take advantage of the structure (1); one of them was trained on-policy and the other one was trained off-policy (see supplement). The evaluation on the test tasks $\mathcal{M}'$ was done by "freezing" the agent at different stages of the learning process and using the GPI policy (3) to select actions. To collect and process the data we used an asynchronous scheme similar to IMPALA (Espeholt et al., 2018).

## 4.3 RESULTS AND DISCUSSION

Fig. 4 shows the results of the agents after being trained on $\mathcal{M}$. One thing that immediately stands out in the figure is the fact that *all* architectures generalise quite well to the test tasks. This is a surprisingly good result when we consider the difficulty of the scenario considered: recall that the agents are solving the test tasks *without any learning taking place*. This performance is even more impressive when we note that some test tasks contain negative rewards, something never experienced by the agents during training. When we look at the relative performance of the agents, it is clear that USFAs perform considerably better than the unstructured UVFAs. This is true even for the case where $\mathcal{C} = \{\mathbf{w}'\}$, in which USFAs essentially reduce to a structured UVFA that was trained by decoupling tasks and policies. The fact that USFAs outperform UVFAs in the scenario considered here is not particularly surprising, since the former exploit the structure (1) while the latter cannot. In any case, it is reassuring to see that our model can indeed exploit such a structure effectively. This result also illustrates a particular way of turning prior knowledge about a problem into a favourable inductive bias in the UVFA architecture.

It is also interesting to see how the different instantiations of USFAs compare against each other. As shown in Fig. 4, there is a clear advantage in including $\mathcal{M}$ to the set of policies $\mathcal{C}$ used in GPI (3). This suggests that, in the specific instantiation of this problem, the type of generalisation provided by SF& GPI is more effective than that associated with UVFAs. One result that may seem counter-intuitive at

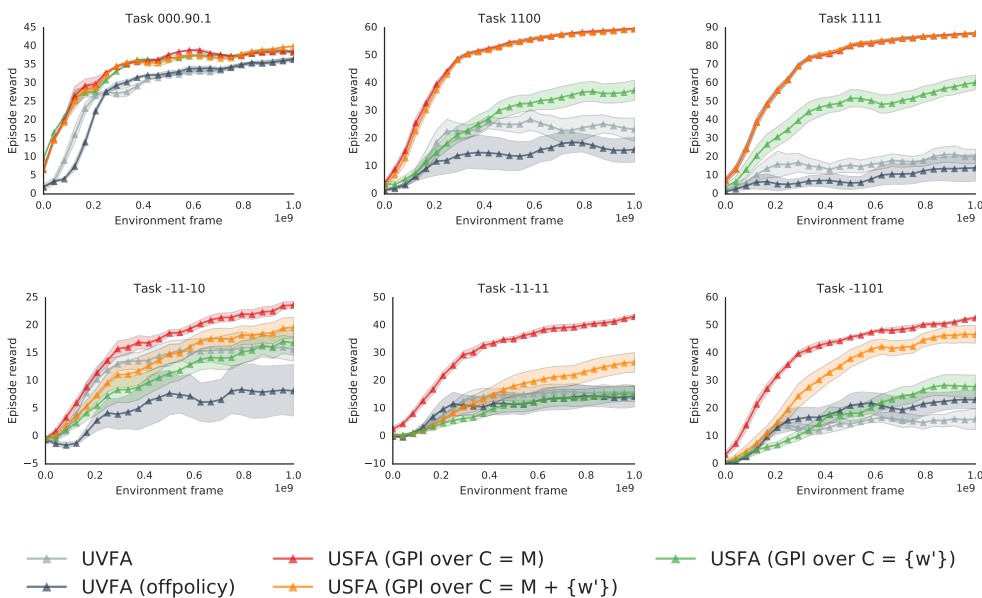

Figure 4: Zero-shot generalisation performance, across different models, on a sample of test tasks $\mathbf{w}' \in \mathcal{M}'$ after training on $\mathcal{M}$. Shaded areas represent one standard deviation over 10 runs.

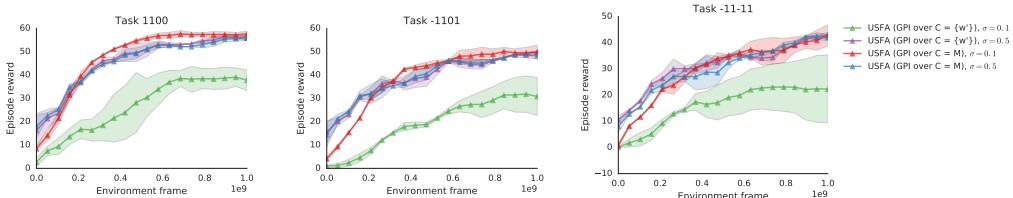

Figure 5: Generalisation performance on sample test tasks $\mathbf{w}' \in \mathcal{M}'$ after training on $\mathcal{M}$, with $\mathcal{D}_{\mathbf{z}} = \mathcal{N}(\mathbf{w}, \sigma \mathbf{I})$, for $\sigma = 0.1$ and $\sigma = 0.5$ (larger coverage of the $\mathbf{z}$ space). Average over 3 runs.

first is the fact that USFAs with $\mathcal{C} = \mathcal{M} + \{\mathbf{w}'\}$ sometimes perform worse than their counterparts using $\mathcal{C} = \mathcal{M}$, especially on tasks with negative rewards. Here we note two points. First, although including more tasks to $\mathcal{C}$ results in stronger guarantees for the GPI policy, strictly speaking there are no guarantees that the resulting policy will perform better (see Barreto et al.'s Theorem 1, 2017). Another explanation very likely in the current scenario is that errors in the approximations $\tilde{\psi}(s, a, \mathbf{z})$ may have a negative impact on the resulting GPI policy (3). As argued previously, adding a point to $\mathcal{C}$ can sometimes increase the upper bound in (4), if the approximation at this point is not reliable. On the other hand, comparing USFA's results using $\mathcal{C} = \mathcal{M} + \{\mathbf{w}'\}$ and $\mathcal{C} = \{\mathbf{w}'\}$, we see that by combining the generalisation of UVFAs and GPI we can boost the performance of a model that only relies on one of them. This highlights the fine balance between the two error terms in (4) and emphasizes how critical selecting low-error candidates in $\mathcal{C}$ can be.

In the above scenario, SF&GPI on the training set $\mathcal{M}$ seems to provide a more effective way of generalising, as compared to UVFAs, even when the latter has a structure specialised to (1). Nevertheless, with less conservative choices of $\mathcal{D}_{\mathbf{z}}$ that provide a greater coverage of the $\mathbf{z}$ space we expect the *structured* UVFA ($C = \{\mathbf{w}'\}$) to generalise better. Note that this can be done without changing $\mathcal{M}$ and is not possible with conventional UVFAs. One of the strengths of USFAs is exactly that: by disentangling tasks and policies, one can learn about the latter without ever having to actually try them out in the environment. We exploit this possibility to repeat our experiments now using $\mathcal{D}_{\mathbf{z}} = \mathcal{N}(\mathbf{w}, 0.5\,\mathbf{I})$. Results are shown in Fig.5. As expected, the generalisation of the structured UVFA improves considerably, almost matching that of GPI. This shows that USFAs can operate in two regimes: i) with limited coverage of the policy space, GPI over $\mathcal{M}$ will provide a reliable generalisation; ii) with a broader coverage of the space structured UVFAs will do increasingly better.[1]

---

[1]Videos of USFAs in action on the links https://youtu.be/Pn76cfXbf2Y and https://youtu.be/0afwHJofbB0.

## 5 RELATED WORK

Multitask RL is an important topic that has generated a large body of literature. Solutions to this problem can result in better performance on the training set (Espeholt et al., 2018), can improve data efficiency (Teh et al., 2017) and enable generalisation to new tasks. For a comprehensive presentation of the subject please see Taylor and Stone (2009) and Lazaric (2012) and references therein.

There exist various techniques that incorporate tasks directly into the definition of the value function for multitask learning (Kaelbling, 1993; Ashar, 1994; Sutton et al., 2011). UVFAs have been used for zero-shot generalisation to combinations of tasks (Mankowitz et al., 2018; Hermann et al., 2017), or to learn a set of fictitious goals previously encountered by the agent (Andrychowicz et al., 2017).

Many recent multitask methods have been developed for learning subtasks or skills for a hierarchical controller (Vezhnevets et al., 2017; Andreas et al., 2016; Oh et al., 2017). In this context, Devin et al. (2017) and Heess et al. (2016) proposed reusing and composing sub-networks that are shared across tasks and agents in order to achieve generalisation to unseen configurations. Finn et al. (2017) uses meta-learning to acquire skills that can be fine-tuned effectively. Sequential learning and how to retain previously learned skills has been the focus of a number of investigations (Kirkpatrick et al., 2016; Rusu et al., 2016). All of these works aim to train an agent (or a sub-module) to generalise across many subtasks. All of these can be great use-cases for USFAs.

USFAs use a UVFA to estimate SFs over multiple policies. The main reason to do so is to apply GPI, which provides a superior zero-shot policy in an unseen task. There have been previous attempts to combine SFs and neural networks, but none of them used GPI (Kulkarni et al., 2016; Zhang et al., 2016). Recently, Ma et al. (2018) have also considered combining SFs and UVFAs. They propose building a goal-conditioned policy that aims to generalise over a collection of goals. In their work, the SFs are trained to track this policy and only used to build the critic employed in training the goal-conditioned policy. Thus, they are considering the extrapolation in $\pi(s, g)$ and using the SFs as an aid in training. Moreover, as both the training and SFs and $\pi(s, g)$ are done on-policy, the proposed system has only seen instances where the SFs, critic and policy are all conditioned on the same goal. In contrast, in this work we argue and show the benefits of decoupling the task and policy to enable generalisation via GPI when appropriate, while preserving the ability to exploit the structure in the policy space. We use the SFs as a way to factorize and exploit effectively the structure in value function space. And we will use these evaluations directly to inform our action selection

## 6 CONCLUSION

In this paper we presented USFAs, a generalisation of UVFAs through SFs. The combination of USFAs and GPI results in a powerful model capable of exploiting the same types of regularity exploited by its precursors: structure in the value function, like UVFAs, and structure in the problem itself, like SF&GPI. This means that USFAs can not only recover their precursors but also provide a whole new spectrum of possible models in between them. We described the choices involved in training and evaluating a USFA and discussed the trade-offs associated with these alternatives. To make the discussion concrete, we presented two examples aimed to illustrate different regimes of operation. The first example embodies a MDP where the generalisation in the optimal policy space is fairly easy but the number of optimal policies we would want to represent can be large. This is a scenario where UVFAs would strive, while vanilla SF&GPI will struggle due to the large number of policies needed to build a good GPI policy. In this case, we show that USFAs can leverage the sort of parametric generalisation provided by UVFAs and even improve on it, due to its decoupled training regime and the use of GPI in areas where the approximation is not quite perfect. Our second example is in some sense a reciprocal one, where we know from previous work that the generalisation provided via GPI can be very effective even on a small set of policies, while generalising in the space of optimal policies, like UVFAs do, seems to require a lot more data. Here we show that USFAs can recover the type of generalisation provided by SFs when appropriate. This example also highlights some of the complexities involved in training at scale and shows how USFAs are readily applicable to this scenario. Overall, we believe USFAs are a powerful model that can exploit the available structure effectively: i) the structure induced by the shared dynamics (via SFs), ii) the structure in the policy space (like UVFAs) and finally iii) the structure in the RL problem itself (via GPI), and could potentially be useful across a wide range of RL applications that exhibit these properties.

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

# Universal Successor Features Approximators
## - Supplementary Material -

## A  TWO TYPES OF GENERALISATION: INTUITION

In this paper we argue that one of the main benefits provided by USFAs is the ability to combine the types of generalisation associated with UVFAs and GPI. In this section, we will take a close look at a very simple example to illustrate the two of generalisation we are considering and how they are different. This is a very small example where the number of optimal policies are very limited and the induced tasks are not that interesting, but we chose this solely to illustrate the decision process induced by GPI and how it differs from parametric generalisation in $w$ via a functional approximator (FA).

Let us consider a an MDP with a single state $s$ and two actions. Upon executing action $a_1$ the agent gets a reward of $0$ and remains in state $s$; the execution of action $a_2$ leads to a potentially non-zero reward followed by termination. We define unidimensional features $\phi \in \mathbb{R}$ as $\phi(s, a_1) = 0$ and $\phi(s, a_2) = 1$. A task is thus induced by a scalar $w \in \mathbb{R}$ which essentially re-scales $\phi(s, a_2)$ and defines the reward $r_w = w$ the agent receives before termination. In this environment, the space of tasks considered are induced by a scalar $\mathbf{w} \in \mathbb{R}$. In this space of tasks, one can easily see that there are only two optimal policies: taking action $a_1$ and receiving the reward $r_{\mathbf{w}} = \mathbf{w}$ if $\mathbf{w} \leq 0$, or taking action $a_0$ and remaining in $s_0$ indefinitely. Thus the space of optimal value functions is very simple. For convenience, we include a depiction of this space in Figure 6.

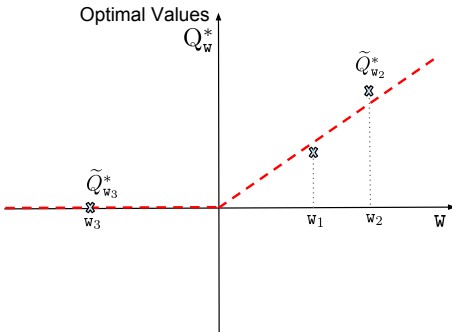

Figure 6: Optimal value space as a function of a scalar task description $\mathbf{w}$

Suppose now we are in the scenario studied in the paper, where after training on a set of tasks $\mathcal{M}$ the agent should generalise to a test task $\mathbf{w}'$. Specifically, let us consider three points in this space $\mathcal{M} = \{\mathbf{w}_1, \mathbf{w}_2, \mathbf{w}_3\}$– three tasks we are going to consider for learning and approximating their optimal policies $\{\tilde{Q}^*_{\mathbf{w}_1}, \tilde{Q}^*_{\mathbf{w}_1}, \tilde{Q}^*_{\mathbf{w}_1}\}$. Given these three points we are going to fit a parametric function that aims to generalise in the space of $\mathbf{w}$. A depiction of this is included in Figure 7(a). Now, given a new point $\mathbf{w}'$ we can obtain a zero-shot estimate $\tilde{Q}^*_{\mathbf{w}'}$ for $Q^*_{\mathbf{w}'}$ – see Figure 7(b). Due to approximation error under a very limited number of training points, this estimate will typically not recover perfectly $Q^*_{\mathbf{w}'} = 0$. In the case of UVFA (and other FAs trying to generalise in task space), we are going to get a guess based on optimal value function we have built, and we are going to take decision based on this estimate $\tilde{Q}^*_{\mathbf{w}'}$.

Given the same base tasks $\mathcal{M} = \{\mathbf{w}_1, \mathbf{w}_2, \mathbf{w}_3\}$ we can now look at what the other method of generalisation would be doing. We are going to denote by $\pi_{z_i}$ the (inferred) optimal policy of task $w_i$. Since we have learnt the SFs corresponding to all of these policies $\psi^{\pi_{z_i}}$, we can now evaluate how well each of these policies would do on the current test task $\mathbf{w}'$: $Q^{\pi_z}_{\mathbf{w}'}(s, a) = \psi^{\pi_z}(s, a)^T \mathbf{w}'$ for all $z \in \mathcal{M}$. A depiction of this step is included in Figure 8(a). Given these evaluations of previous behaviours, GPI takes the maximum of these values – "trusting", in a sense, the most promising value. In our case this corresponds to the behaviour associated with task $\mathbf{w}_3$, which in this case happens to

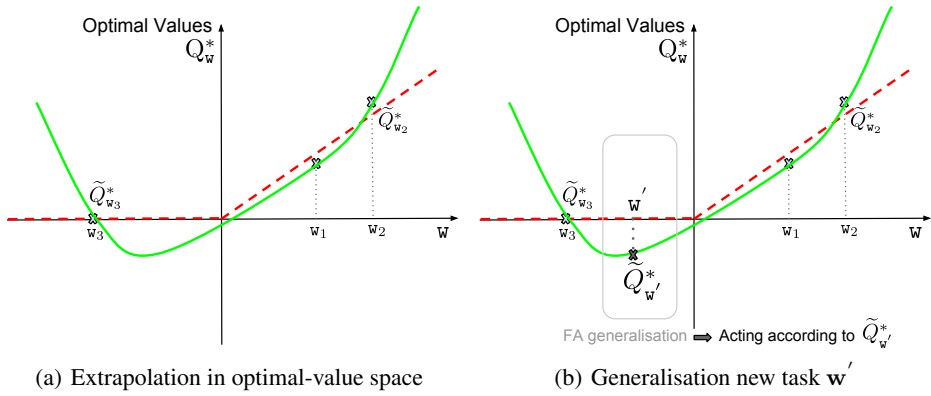

Figure 7: UVFA-like generalisation.

have the same optimal policy as our test task $\mathbf{w}'$. Thus in this particular example, the evaluation of a previously learned behaviour gives us a much better basis for inducing a behaviour in our test task $\mathbf{w}'$. Moreover if the SFs are perfect we would automatically get the optimal value function for $\mathbf{w}'$.

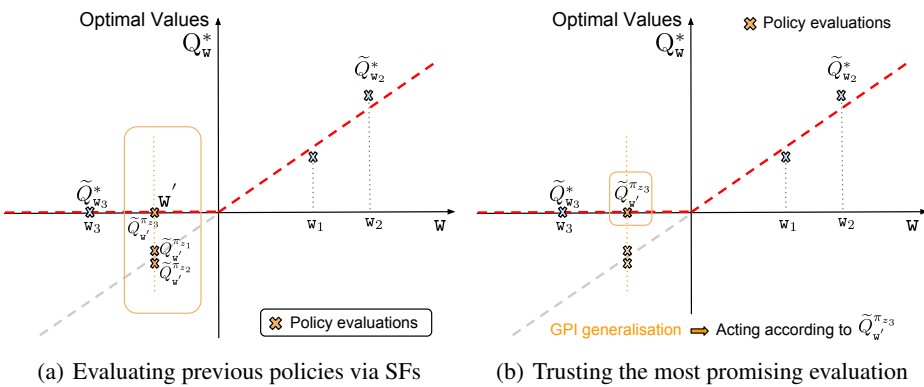

Figure 8: GPI generalisation.

It is worth noting that, in this case, by learning a USFA we can recover both scenarios described above based on our choice of the candidate set for GPI $\mathcal{C}$. In particular, if $\mathcal{C} = \{\mathbf{w}'\}$ we recover the mechanism in Figure 7, while for $\mathcal{C} = \mathcal{M}$ we recover the generalisation in Figure 8. Furthermore, for any choice of *trained* points that include one positive and one negative choice of $\mathbf{w}$, by relying on GPI we can generalise perfectly to the whole space of tasks, while an approach based exclusively on the sort of generalisation provided by UVFAs may struggle to fit the full function. Analogously, in scenarios where the structure of the optimal space favours (UV)FAs, we expect USFAs to leverage this type of generalisation. An example of such a scenario is given in the first part of the experimental section – Section 4.1, and further details in Section B.

# B  ILLUSTRATIVE EXAMPLE: TRIP MDP

In this section we provide some additional analysis and results omitted from the main text. As a reminder, this is a two state MDP, where the first state is a root state, the transition from $s_1 \to s_2$ comes at a cost $r_{\mathbf{w}}(s_1, E) = \phi(s_1, E)^T \mathbf{w} = -\epsilon(w_1 + w_2)$ and all other actions lead to a final positive reward corresponding to how much the resulting state/restaurant alligns with our preferences (our task) right now. For convenience, we provide below the depiction of the Trip MDP introduced in Section 4.1.

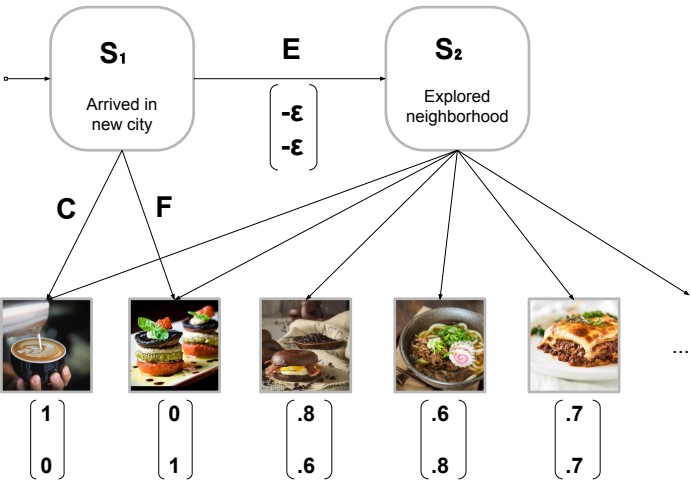

In the experiments run we considered a fixed set of training tasks $\mathcal{M} = \{01, 10\}$ for all methods. The set of outcomes from the exploratory state $s_2$ is defined as $\phi(s_2, a) = [\cos(\theta), \sin(\theta)]$ for $\theta \in \{k\pi/2N\}_{k=0,N}$. Note that this includes the binary states for $k = 0$ and respectively $k = N$. We ran this MDP with $N = 6$, and $\epsilon = 0.05$. Thus outside the binary outcomes, the agent can select $N - 1 = 5$ other mixed outcomes and, as argued in the main text, under these conditions there will be a selection of the $\mathbf{w}$-space in which each of these outcomes will be optimal. Thus the space of optimal policies, we hope to recover, is generally $N + 1$. Nevertheless, there is a lot of structure in this space, that the functional approximators can uncover and employ in their generalization.

## B.1  ADDITIONAL RESULTS

In the main paper, we reported the zero-shot aggregated performance over all direction $\mathcal{M}' = \{\mathbf{w}' | \mathbf{w}' = [\cos(\frac{\pi k}{2K}), \sin(\frac{\pi k}{2K})], k \in \mathbb{Z}_K\}$. This should cover most of the space of tasks/trade-offs we would be interest in. In this section we include the generalization for other sets $\mathcal{M}'$. First in Fig. 9 we depict the performance of the algorithms considered across the whole $\mathbf{w}'$ space $\mathcal{M}' = [0, 1]^2$. Fig. 10 is just a different visualization of the previous plot, where we focus on how far these algorithms are from recovering the optimal performance. This also shows the subtle effect mentioned in the discussion in the main text, induced by the choice of $\mathcal{C}$ in the USFA evaluation.

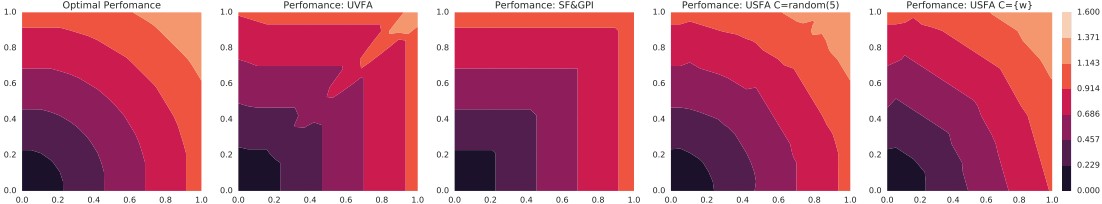

Figure 9: [Sample run] Performance of the different methods (in this order, starting with the second subplot): UVFA, SF&GPIon the perfect SFs induced by $\mathcal{M}$, USFA with $\mathcal{C} = random(5)$ and USFA with $\mathcal{C} = \{\mathbf{w}'\}$ as compared to the optimal performance one could get in this MDP (first plot). These correspond to one sample run, where we trained the UVFA and USFA for 1000 episodes. The optimal performance and the SF&GPIwere computed exactly.

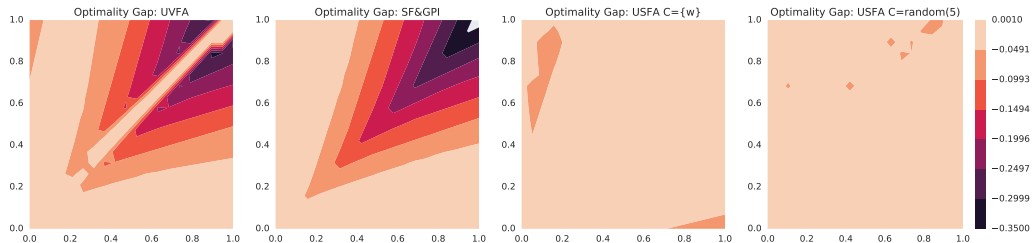

Figure 10: [Sample run] Optimality gap over the whole task space. These correspond to the same sample run as above, where we trained the UVFA and USFA for 1000 episodes. We can now see more clearly that USFAs manage to recover better policies and optimality across a much greater portion of the task space. The last two plots correspond to the same USFA just using different choices of the candidate set $\mathcal{C}$. Something to note here is that by having a more diverse choice in $\mathcal{C}$, we can recover an optimal policy even in areas of the space where our approximation has not yet optimally generalised (like the upper-left corner in the $\mathbf{w}$-space in the figures above).

A particularly adversarial choice of test tasks for the vanilla SF&GPIwould be the diagonal in the $[0,1]^2$ quadrant depicted in the plot above: $\mathcal{M}' = \{\mathbf{w}'|w_1' = w_2', w_1 \in [0,1]\}$. This is, in a sense, maximally away from the training tasks and both of the precursor models are bound to struggle in this portion of the space. This intuition was indeed empirically validated. Results are provided in Fig. 11. As mentioned above, this is an adversarial evaluation, mainly to point out that, in general, there might be regions of the space were the generalization of the previous models can be very bad, but where the combination of them can still recover close to optimal performance.

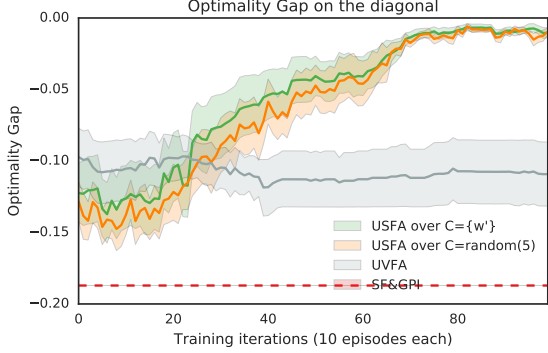

Figure 11: Zero-shot performance on the diagonal: Optimality gap for $\mathcal{M}' = \{\mathbf{w}'|w_1' = w_2', w_1 \in [0,1]\}$. These results were averaged over 10 runs.

## C  LARGE SCALE EXPERIMENTS: DETAILS

### C.1  AGENT'S ARCHITECTURE

This section contains a detailed description of the USFA agent used in our experimental section. As a reminder, we include the agent's architecture below (Figure 1 in the main text).

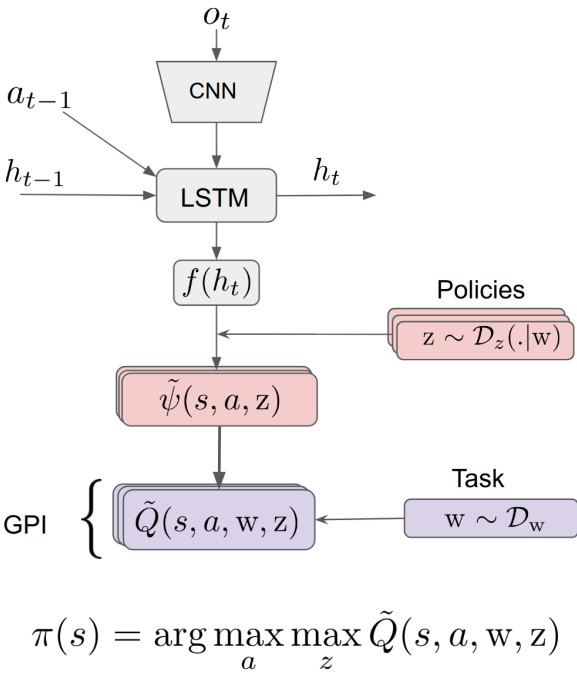

$$\pi(s) = \arg \max_a \max_z \tilde{Q}(s, a, \mathrm{w}, \mathrm{z})$$

Figure 12: USFA architecture

As highlighted in Section 4.2, our agent comprises of three main modules:

- **Input processing module**: computes a state representation $f(h_t)$ from observation $o_t$. This module is made up of three convolutional layers (structure identical to the one used in (Mnih et al., 2015)), the output of which then serves as input to a LSTM (256). This LSTM takes as input the previously executed action $a_{t-1}$. The output of the LSTM is passed through a non-linearity $f$ (chosen here to be a ReLu) to produce a vector of 128 units, $f(h_t)$.

- **Policy conditioning module:** compute the SFs $\tilde{\psi}(s, a, \mathbf{z})$, given a (sampled) policy embedding $\mathbf{z}$ and the state representation $f(h_t)$. This module first produces $n_{\mathbf{z}}$ number of $\mathbf{z} \sim \mathcal{D}_{\mathbf{z}}$ samples ($n_{\mathbf{z}} = 30$ in our experiments). Each of these is then transformed via a 2-layer MLP(32,32) to produce a vector of size 32, for each sample $\mathbf{z}$. This vector $g(\mathbf{z})$ gets concatenated with the state representation $f(h_t)$ and the resulting vector is further processed by a 2-layer MLP that produces a tensor of dimensions $d \times |\mathcal{A}|$ for each $\mathbf{z}$, where $d = dim(\phi)$. These correspond to SFs $\tilde{\psi}(s, a, \mathbf{z})$ for policy $\pi_{\mathbf{z}}$. Note that this computation can be done quite efficiently by reusing the state embedding $f(h_t)$, doing the downstream computation in parallel for each policy embedding $\mathbf{z}$.

- **Task evaluation module:** computes the value function $Q(s, a, \mathbf{z}, \mathbf{w}) = \tilde{\psi}(s, a, \mathbf{z})^T \mathbf{w}$ for a given task description $\mathbf{w}$. This module does not have any parameters as the value functions are simply composable from $\tilde{\psi}(s, a, \mathbf{z})$ and the task description $\mathbf{w}$ via assumption (1). This module with output $n_{\mathbf{z}}$ value functions that will be used to produce a behavior via GPI.

An important decision in this design was how and where to introduce the conditioning on the policy. In all experiments shown here the conditioning was done simply by concatenating the two embeddings $\mathbf{w}$ and $\mathbf{z}$, although stronger conditioning via an inner product was tried yielding similar performance.

The 'where' on the other hand is much more important. As the conditioning on the policy happens quite late in the network, most of the processing (up to $f(h_t)$) can be done only once, and we can sample multiple $\mathbf{z}$ and compute the corresponding $\tilde{\psi}^{\pi_{\mathbf{z}}}$ at a fairly low computational cost. As mentioned above, these will be combined with the task vector $\mathbf{w}$ to produce the candidate action value functions for GPI. Note that this helps both in training and in acting, as otherwise the unroll of the LSTM would be policy conditioned, making the computation of the SFs and the off-policy $n$-step learning quite expensive. Furthermore, if we look at the learning step we see that this step can also benefit from this structure, as the gradient computation of $f$ can be reused. We will only have a linear dependence on $n_z$ on the update of the parameters and computations in the red blocks in Figure 12.

UVFA baseline agents have a similar architecture, but now the task description $\mathbf{w}$ is fed in as an input to the network. The conditioning on the task of UVFAs is done in a similar fashion as we did the conditioning on the policies in USFAs, to make the computational power and capacity comparable. The input processing module is the same and now downstream, instead of conditioning on the policy embedding $\mathbf{z}$, we condition on task description $\mathbf{w}$. This conditioning if followed by a 2-layer MLP that computes the value functions $\tilde{Q}^*(s, a, \mathbf{w})$, which induces the greedy policy $\pi_{\mathbf{w}}^{(UVFA)} = \arg\max_a \tilde{Q}^*(s, a, \mathbf{w})$.

### C.2    AGENT'S TRAINING

The agents' training was carried out using the IMPALA architecture (Espeholt et al., 2018). On the learner side, we adopted a simplified version of IMPALA that uses $Q(\lambda)$ as the RL algorithm. In our experiments, for all agents we used $\lambda = 0.9$. Depending on the sampling distribution $D_{\mathbf{z}}$, in learning we will be often off-policy. That is, most of the time, we are going to learn about a policy $\pi_{\mathbf{z}_1}$ and update its corresponding SFs approximations $\tilde{\psi}(s, a, \mathbf{z}_1)$, using data generated by acting in the environment according to some other policy $\pi_{\mathbf{z}_2}$. In order to account for this off-policiness, whenever computing the n-step return required in eq. 5, we are going to cut traces whenever the policies start to disagree and bootstrap from this step on (Sutton and Barto, 1998). Here we can see how the data distribution induce by the choice of training tasks $\mathcal{M}$ can influence the training process. If the data distribution $\mathcal{D}_{\mathbf{z}}$ is very close to the set $\mathcal{M}$, as in our first experiment, most of the policies we are going to sample will be close to the policies that generated the data. This means that we might be able to make use of longer trajectories in this data, as the policies will rarely disagree. On the other hand, by staying close to the training tasks, we might hurt our ability to generalise in the policy space, as our first experiment suggest (see Figure 4). By having a broader distribution $\mathcal{D}_z = \mathcal{N}(\mathbf{w}, 0.5I)$, we can learn about more diverse policies in this space, but we will also increase our off-policiness. We can see from Figure 5, that our algorithm can successfully learn and operate in both of these regimes.

For the distributed collection of data we used 50 actors per task. Each actor gathered trajectories of length 32 that were then added to the common queue. The collection of data followed an $\epsilon$-greedy policy with a fixed $\epsilon = 0.1$. The training curves shown in the paper correspond to the performance of the the $\epsilon$-greedy policy (that is, they include exploratory actions of the agents).

### C.3    AGENT'S EVALUATION

All agents were evaluated in the same fashion. During the training process, periodically (every 20M frames) we will evaluate the agents performance on a test of held out test tasks. We take these intermediate snapshots of our agents and 'freeze' their parameters to assess zero-shot generalisation. Once a test task $\mathbf{w}'$ is provided, the agent interacts with the environment for 20 episodes, one minute each and the average (undiscounted) reward is recorded. These produce the evaluation curves in Figure 4. Evaluations are done with a small $\epsilon = 0.001$, following a GPI policy with different instantiations of $\mathcal{C}$. For the pure UVFA agents, the evaluation is similar: $\epsilon$-greedy on the produced value functions $\tilde{Q}^*(s, a, \mathbf{w})$, with the same evaluation $\epsilon = 0.001$.

## D    ADDITIONAL RESULTS

In our experiments we defined a set of easy test tasks (close to $\mathcal{M}$) and a set of harder tasks, in order to cover reasonably well a few distinct scenarios:

- Testing generalisation to tasks very similar to the training set, e.g. $\mathbf{w}' = [0, 0.9, 0, 0.1]$;
- Testing generalisation to harder tasks with different reward profiles: only positive rewards, only negative rewards, and mixed rewards.

In the main text, we included only a selection of these for illustrative purposes. Here we present the full results.

### D.1 CANONICAL BASIS: ZERO-SHOT GENERALISATION

This section contains the complete results of the first experiment conducted. As a reminder, in this experiment we were training a USFA agent on $\mathcal{M} = \{1000, 0100, 0010, 0001\}$, with $D_{\mathbf{z}} = \mathcal{N}(\mathbf{w}, 0.1I)$ and compare its performance with two conventional UVFA agents (one trained on-policy and the other one using all the data generated to learn off-policy) on a range of unseen test tasks. Complete set of result is included below, as follows: Figure 13 includes results on easy tasks, close to the tasks contained in the training set $\mathcal{M}$ (generalisation to those should be fairly straightforward); Figure 14 and Figure 15 present results on more challenging tasks, quite far away from the training set, testing out agents ability to generate to the whole 4D hypercube.

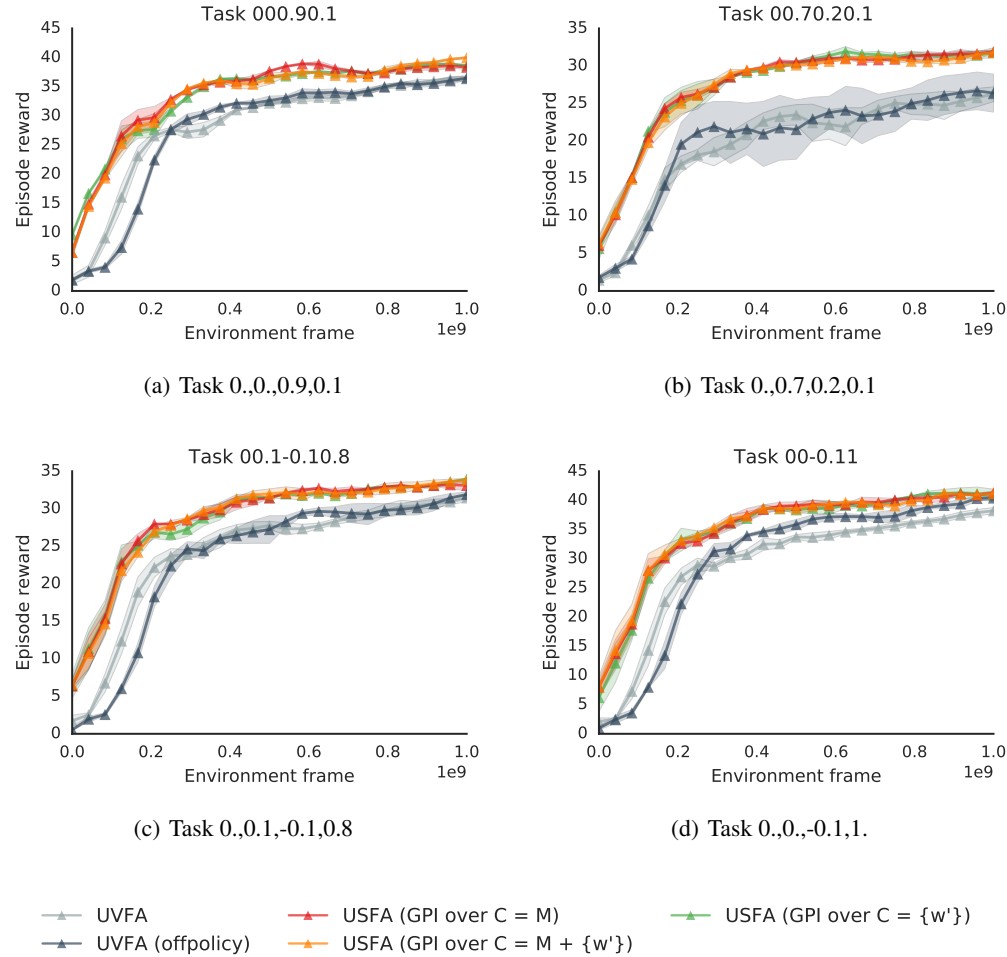

Figure 13: **Zero-shot performance on the easy evaluation set**: Average reward per episode on test tasks not shown in the main paper. This is comparing a USFA agent trained on the canonical training set $\mathcal{M} = \{1000, 0100, 0010, 0001\}$, with $D_{\mathbf{z}} = \mathcal{N}(\mathbf{w}, 0.1I)$ and the two UVFA agents: one trained on-policy, one employing off-policy.

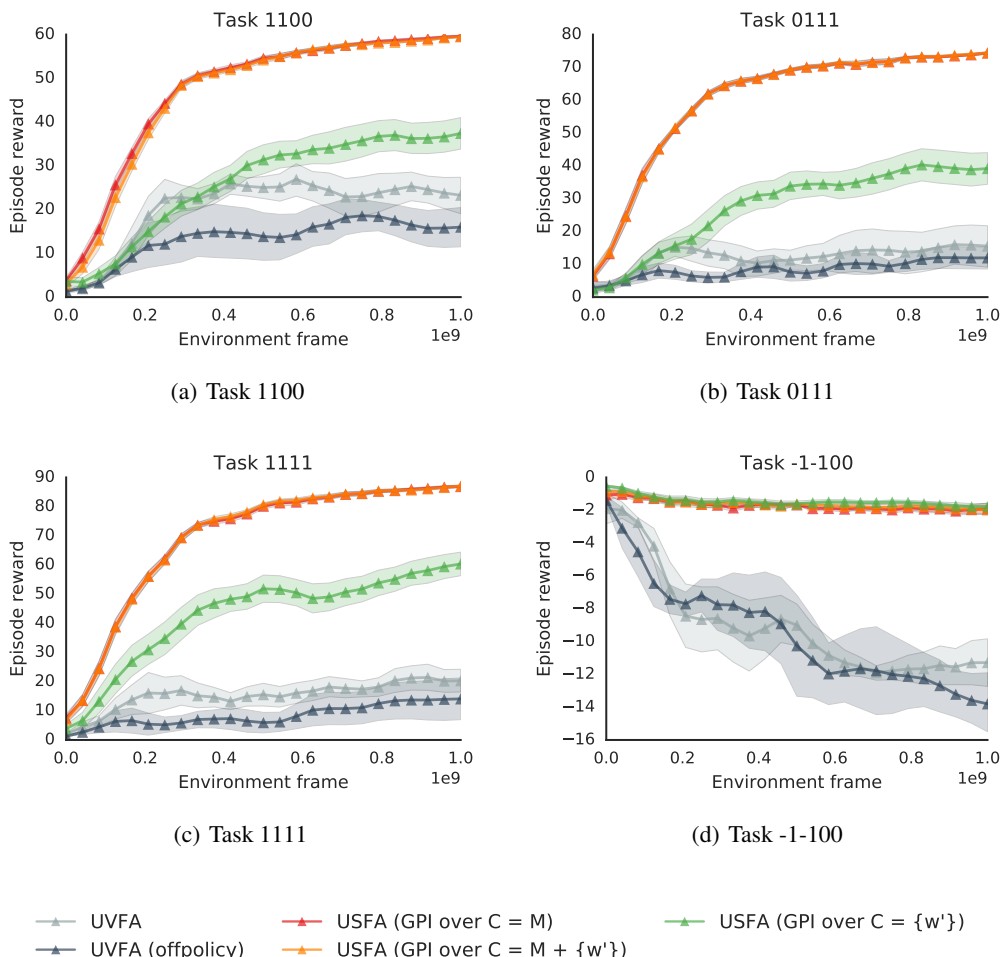

(a) Task 1100

(b) Task 0111

(c) Task 1111

(d) Task -1-100

Figure 14: **Zero-shot performance on harder tasks**: Average reward per episode on test tasks not shown in the main paper. This is comparing a USFA agent trained on the canonical training set $\mathcal{M} = \{1000, 0100, 0010, 0001\}$, with $D_{\mathbf{z}} = \mathcal{N}(\mathbf{w}, 0.1I)$ and the two UVFA agents: one trained on-policy, one employing off-policy. (Part 1)

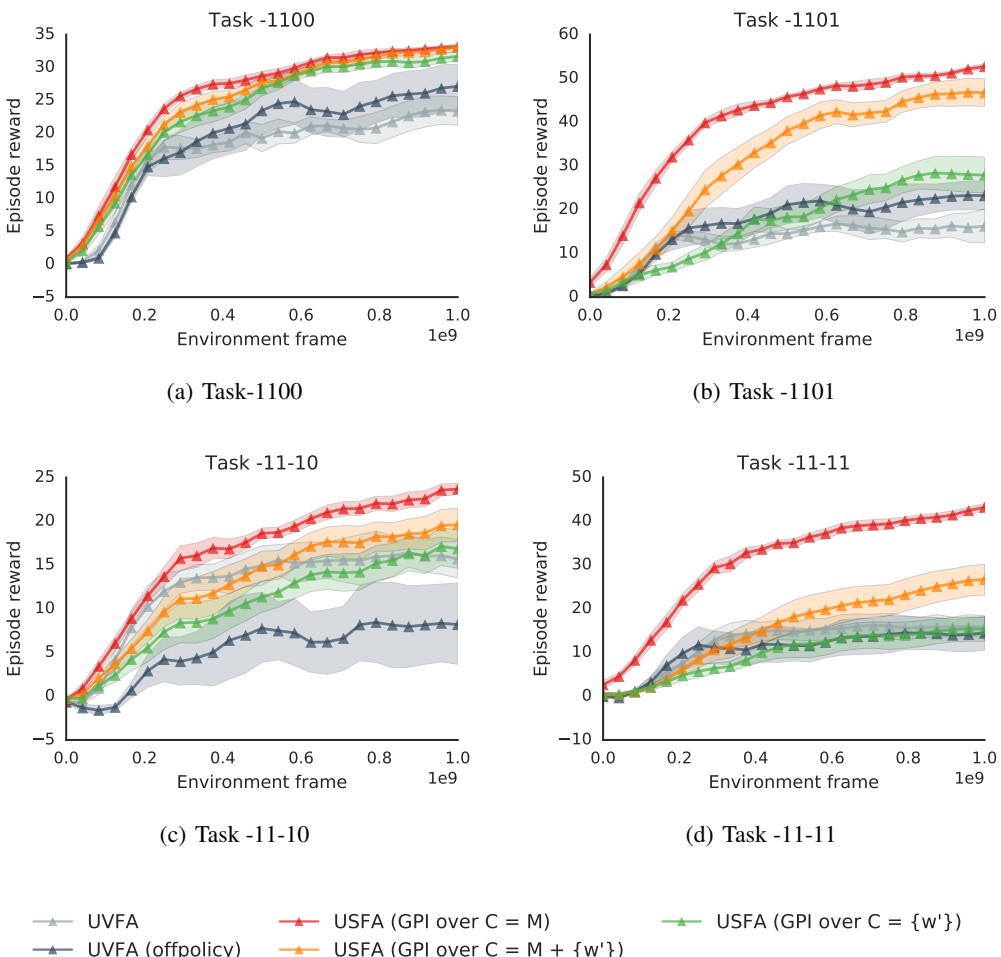

(a) Task-1100

(b) Task -1101

(c) Task -11-10

(d) Task -11-11

Figure 15: **Zero-shot performance on harder tasks**: Average reward per episode on test tasks not shown in the main paper. This is comparing a USFA agent trained on the canonical training set $\mathcal{M} = \{1000, 0100, 0010, 0001\}$, with $D_{\mathbf{z}} = \mathcal{N}(\mathbf{w}, 0.1I)$ and the two UVFA agents: one trained on-policy, one employing off-policy. (Part 2)

### D.2 CANONICAL BASIS: USFAS IN DIFFERENT TRAINING REGIMES.

In this section, we include the omitted results from our second experiment. As a reminder, in this experiment we were training two USFA agents on the same set of canonical tasks, but employing different distributions $D_{\mathbf{z}}$, one will low variance $\sigma = 0.1$, focusing in learning policies around the training set $\mathcal{M}$, and another one with larger variance $\sigma = 0.5$, that will try to learn about a lot more policies away from the training set, thus potentially facilitating the generalisation provided by the UVFA component. Results are displayed in Figures 16-17 on all tasks in the hard evaluation set.

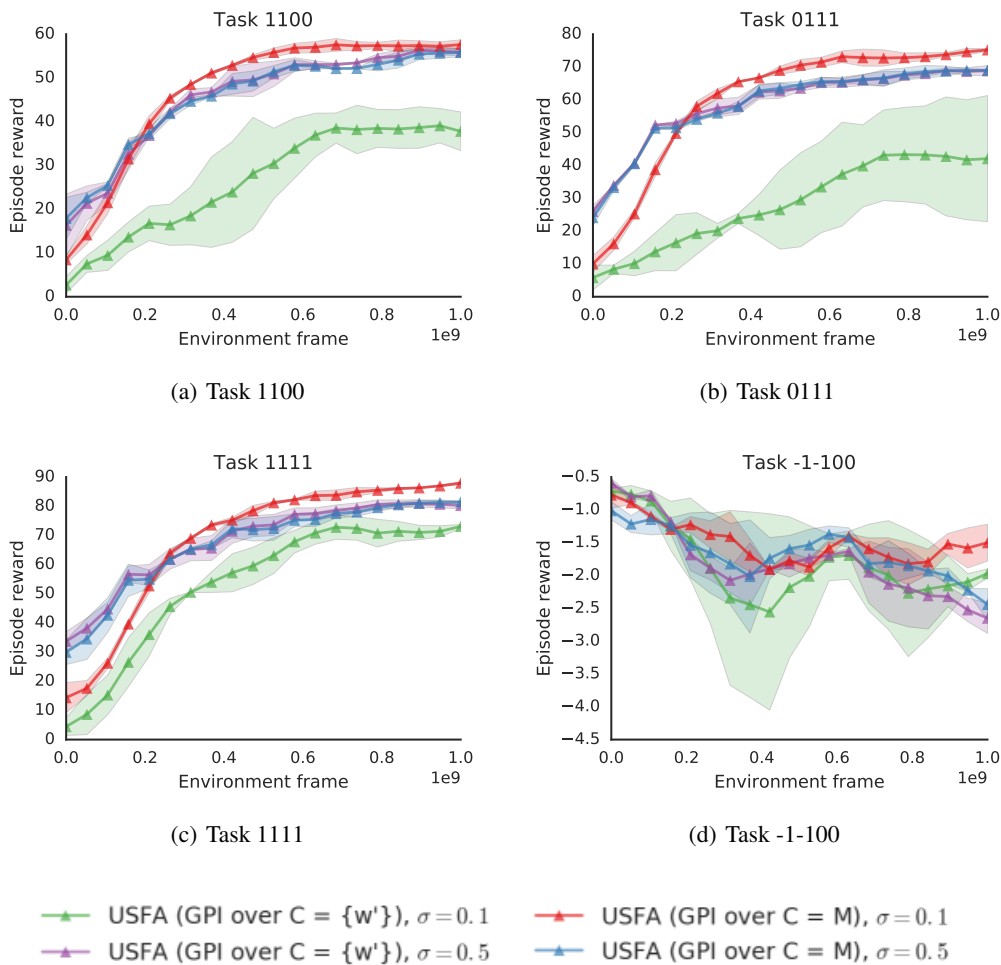

Figure 16: **Different $\mathcal{D}_{\mathbf{z}}$ – Zero-shot performance on harder tasks**: Average reward per episode on test tasks not shown in the main paper. This is comparing the generalisations of two USFA agent trained on the canonical training set $\mathcal{M} = \{1000, 0100, 0010, 0001\}$, with $D_{\mathbf{z}} = \mathcal{N}(\mathbf{w}, 0.1I)$, and $D_{\mathbf{z}} = \mathcal{N}(\mathbf{w}, 0.5I)$. (Part 1)

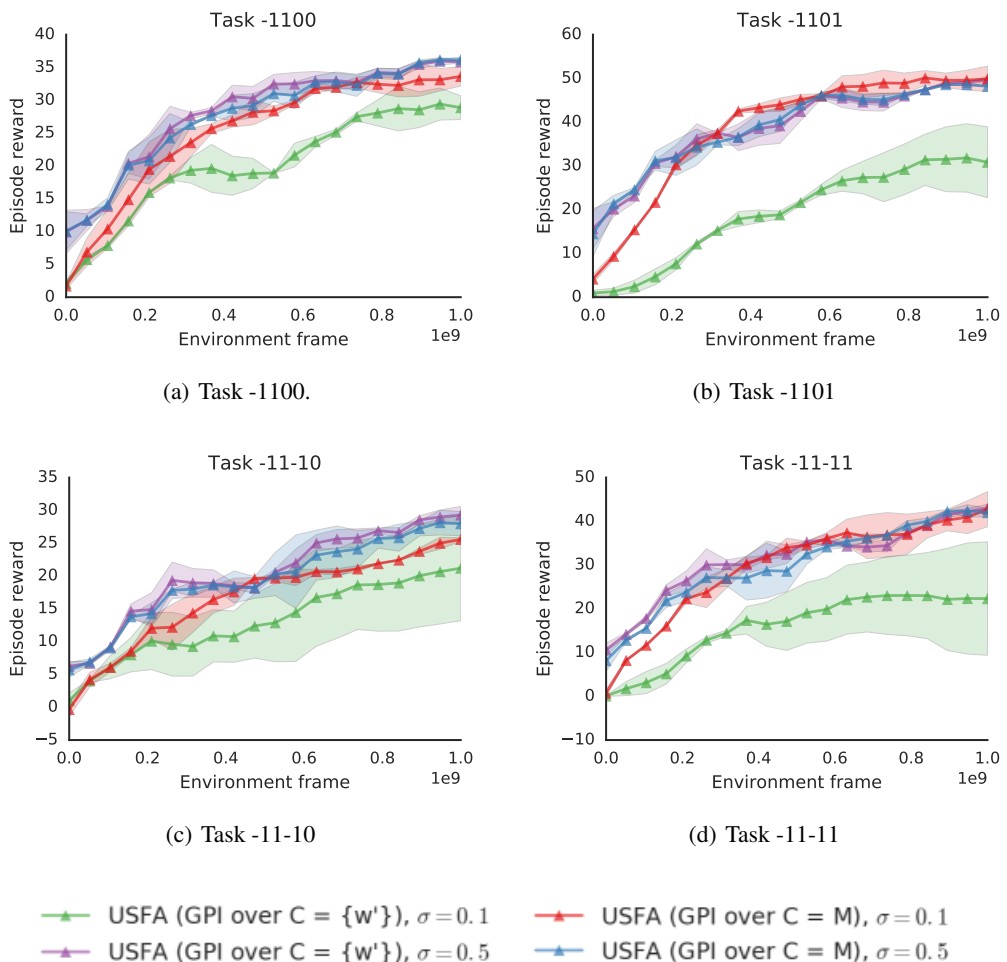

Figure 17: **Different $\mathcal{D}_{\mathbf{z}}$ – Zero-shot performance on harder tasks**: Average reward per episode on test tasks not shown in the main paper. This is comparing a USFA agent trained on the canonical training set $\mathcal{M} = \{1000, 0100, 0010, 0001\}$, with $D_{\mathbf{z}} = \mathcal{N}(\mathbf{w}, 0.1I)$, and $D_{\mathbf{z}} = \mathcal{N}(\mathbf{w}, 0.5I)$. (Part 2)

### D.3 LARGER COLLECTION OF TRAINING TASKS

We also trained our USFA agent on a larger set of training tasks that include the previous canonical tasks, as well as four other tasks that contain both positive and negative reward $\mathcal{M} = \{1000, 0100, 0010, 0001, 1\text{-}100, 01\text{-}10, 001\text{-}1, \text{-}1000\}$. Thus we expect this agent to generalises better as a result of its training. A selection of these results and sample performance in training are included in Fig. 18.

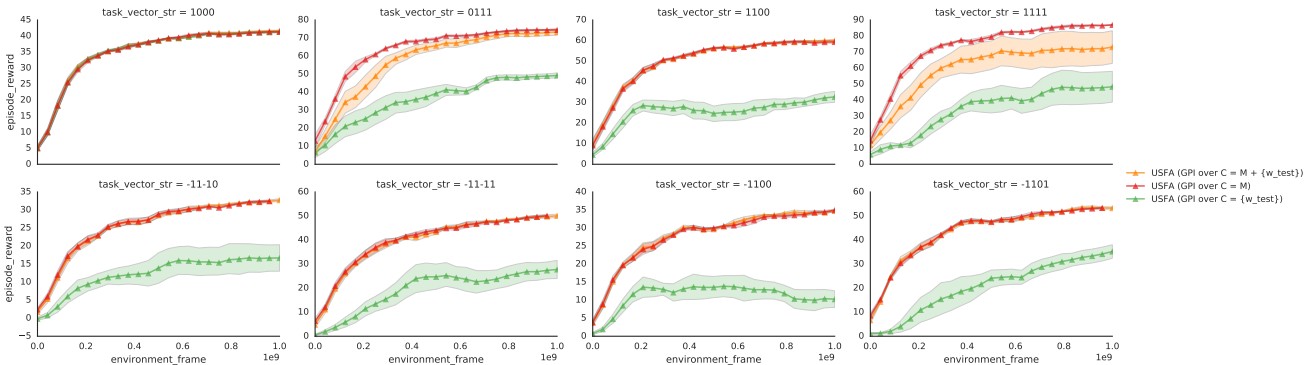

Figure 18: **Large $\mathcal{M}$.** Learning curves for training task $[1000] \in \mathcal{M}$ and generalisation performance on a sample of test tasks $\mathbf{w}' \in \mathcal{M}'$ after training on all the tasks $\mathcal{M}$. This is a selection of the hard evaluation tasks. Results are average over 10 training runs.

