# OpenReview forum: "Universal Successor Features Approximators"
_ICLR.cc/2019/Conference_

### Official Review · AnonReviewer2 · 2018-11-01
**New method for RL generalization to new tasks**

**Rating:** 6
**Confidence:** 4

**Review:**

Paper’s contributions:
This paper considers the challenging problem of generalizing well to new RL tasks, based on having learned on a set of previous related RL tasks.  It considers tasks that differ only in their reward function (assume the dynamics are identical), and where the reward functions are constrained to be linear combinations over a set of given features.  The main approach, Universal Successor Features Approximators (USFAs) is a combination of two recent approaches:  Universal Value Function Approximators (UVFAs) and Generalized Policy Improvement (GPI).  The main claim is that while each of these methods leverages different types of regularity when generalizing to new tasks, USFAs are able to jointly leverage both types (and elegantly have both other methods as special cases).

Summary of evaluation:
Overall the paper tackles an important problem, and provides careful explanation and reasonably extensive results showing the ability of USFA to leverage structure.  I’m on the fence because I really wish the combination of generalization properties could be understood in a more intuitive way.  There are some more minor issues, such as lack of complexity analysis and a few notation details, that can be easily fixed.

Pros:
-	The problem of generalizing to new tasks in RL is an important open problem.
-	The paper is carefully written and provides clear explanation of most of the methods & results.

Cons:
-	The authors are diligent about trying to explain what type of regularities are exploited by each of UVFAs and GPI, and how this can be combined in USFAs.  However despite reading these parts carefully, I could not get a really good intuition, either in the methods or in the results, for the nature of the regularities exploited, and how it really differs.  Top of p.4 says that GPI generalizes well when the policy \pi(s) does well on task w’.  Can you give a specific MDP where Q is not smooth, but the policy does well?
-	There is no complexity analysis.  I would like to know the computational complexity of each of the key steps in Algorithm 1 (with comparison to simple UVFA and GPI).
-	It would be useful to see the empirical comparison with the approach of Ma et al. (2018), which also combines SFs and UFVAs. I understand there are differences in the details, but I would like to see confirmation of whether the claims about USFA’s superior ability to exploit structure is supported by results.

Minor comments:
-	The limitation to linear rewards is a reasonably strong assumption.  It would be good to support this, e.g. by references to domain that meet this assumption.
-	It seems the mathematical properties in Sec.3.1 could be further developed.
-	P.4: “Given a deterministic policy \pi, one can easily define a reward function r_\pi”.  I did not think this mapping was unique (see the literature on IRL, e.g. Ross et al.).  Can you provide a proof or reference to support this statement?
-	The definition of Q(s,a,w,z) is interesting. Can this be seen as a kernel between w and z?
-	\theta suddenly shows up in Algorithm 1. I presume these are the parameters of Q?  Should be defined.
-	The distribution used to sample policies seems to be a key step of this approach, yet not much guidance is given on how to do this in general.

---

### Official Review · AnonReviewer3 · 2018-11-04
**Universal Successor Feature Approximators**

**Rating:** 5
**Confidence:** 2

**Review:**

This paper proposes new ideas in the context of deep multi-task learning for RL. Ideas seem to me to be a rather small (epsiilon) improvement over the cited works.

The main problem - to me - with described approach is that the Q* value now lives in a much higher dimensional space, levelling any advantage a subsequent heuristic might give.

Statements as 'Although this work is superficially similar to ours, it differs a lot in the details' makes clear that this work is only of potential interest for a rather small audience, a tenet also supported by the density of presentation. I leave it to the AC to decide on relevance.

---

### Official Review · AnonReviewer1 · 2018-11-12
**Scheme to generalize Q values across policies and tasks, by combining universal value function approximation and successor features+generalized policy improvement**

**Rating:** 7
**Confidence:** 3

**Review:**

The goal here is multi-task learning and generalization, assuming that the expected one-step reward for any member of the task family can be written as $\phi(s,a,s')^T w$. The authors propose universal successor features (USF) $\psi$s, such that the action-value functions Q can be written as $Q(s,a,w,z)=\psi(s,a,z)^T w$, generalizing over mutiple tasks each denoted by $w$, and multiple policies each denoted by $z$. Here, $z$ represents the optimal policy induced by a reward specified by $z$ (from the same set as $w$). Using USFs $\psi$-s, the Q values can be interpolated across policies and tasks. Due to the disentangling of reward and policy generalizations, the training sets for $w$ and $z$ can be independently sampled. The authors further generalize a temporal difference error in these USFs $\psi$s, using the TD error to learn to approximate the $\psi$s by a network (USF Approximator i.e USFA). They then test the generalization capabilities of these USFAs on families of a simple task and a DeepMind Lab based task.

I find this paper a good fit for ICLR as the paper significantly advances learning representations for Q values that generalize across policies and tasks.

Some issues to consider:
1. Given a policy, I would think that the reward function that induces this policy is not unique. This non-uniqueness probably doesn't matter for the USF development, since the policies are restricted to those induced by z-s (from the same set as w-s), but the authors should clarify this point.

2. I suppose there are no convergence guarantees on the $\psi$-learning?

3. I do believe that this work goes reasonably beyond the Ma et al 2018 paper, and the authors do clarify their advance especially in incorporating generalized policy improvement. However, the authors way of writing makes it appear as if their work only differs in some details. I recommend to remove this unexplanatory sentence:
"Although this work is superficially similar to ours, it differs a lot in the details."

Minor:
page 3: last but one line: "more clear" --> "clearer"
page 3: "In contrast" --> "By contrast" -- but this is not a hard rule

---

### Meta-Review · Area_Chair1 · 2018-12-19
**Nice contribution to multi-task (multiple reward functions) setting for RL**

**Confidence:** 4
**Recommendation:** Accept (Poster)

**Metareview:**

This paper addresses an importnant and more realistic setting of multi-task RL where the reward function changes; the approach is elegant, and empirical results are convincing. The paper presents an importnant contribution to the challenging multi-task RL problem.